# Identification of Bird Habitat Restoration Priorities in a Central Area of a Megacity

**Yuncai Wang** [1,2,*] , **Xinghao Lu** [1], **Ruojing Wang** [1], **Yifei Jia** [1] and **Junda Huang** [1]

1 Department of Landscape Architecture, College of Architecture and Urban Planning, Tongji University, Shanghai 200092, China; kaxingxing1210@tongji.edu.cn (X.L.); 2130158@tongji.edu.cn (R.W.); 2010268@tongji.edu.cn (Y.J.); 1810172@tongji.edu.cn (J.H.)
2 Center of Ecological Planning and Environment Effects Research, Key Laboratory of Ecology and Energy-Saving Study of Dense Habitat, Ministry of Education, Shanghai 200092, China
* Correspondence: wyc1967@tongji.edu.cn; Tel.: +86-021-65980253

**Abstract:** Rapid global urbanization has caused habitat degradation and fragmentation, resulting in biodiversity loss and the homogenization of urban species. Birds play a crucial role as biodiversity indicators in urban environments, providing multiple ecosystem services and demonstrating sensitivity to changes in habitat. However, construction activities often disrupt urban bird habitats, leading to a decline in habitat quality. This paper proposes a framework for prioritizing habitat restoration by pinpointing bird hotspots that demand attention and considering the matching relationship between bird richness and habitat quality. Shanghai represents a typical example of the high-density megacities in China, posing a significant challenge for biodiversity conservation efforts. Utilizing the random forest (RF) model, bird richness patterns in central Shanghai were mapped, and bird hotspots were identified by calculating local spatial autocorrelation indices. From this, the habitat quality of hotspot areas was evaluated, and the restoration priority of bird habitats was determined by matching bird richness with habitat quality through *z*-score standardization. The results were as follows: (1) Outer-ring green spaces, large urban parks, and green areas along coasts or rivers were found to be the most important habitats for bird richness. Notably, forests emerged as a crucial habitat, with approximately 50.68% of the forested areas identified as hotspots. (2) Four habitat restoration types were identified. The high-bird-richness–low-habitat-quality area (HBR-LHQ), mainly consisting of grassland and urban construction land, was identified as a key priority for restoration due to its vulnerability to human activities. (3) The Landscape Shannon's Diversity Index (SHDI) and Normalized Difference Vegetation Index (NDVI) are considered the most significant factors influencing the bird distribution. Our findings provide a scientifically effective framework for identifying habitat restoration priorities in high-density urban areas.

**Keywords:** urban biodiversity; restoration priorities; bird hotspots; habitat quality; random forest model



## 1. Introduction

Urbanization is a major threat to global biodiversity, with 12% of birds, 23% of mammals, and 32% of amphibians on the brink of extinction, according to the Millennium Ecosystem Assessment [1,2]. Moreover, research predicts that the global urban area will increase by 1.8 to 5.9 times by the end of this century [3]. China, specifically, has experienced a rapid acceleration in urbanization since the implementation of reforms and the opening-up policy, with projections indicating that China's urban population will exceed 1 billion by 2030 [4]. Between 1978 and 2020, the total area of urban construction land in China surged from 7438 km² to an expansive 58,355 km² [5]. The consequences of rapid urban expansion include the fragmentation of urban landscapes, resulting in a reduced habitat area and impeded species flow, ultimately leading to the homogenization of urban species [6]. Anthropogenic activities and vegetation loss during the process of urbanization

are the main causes of urban habitat degradation, leading to the loss of survival resources for regional biota and disrupting the ecological balance [7]. The restoration of urban habitats to maintain urban biodiversity is urgently needed. However, due to limited resources, urban decision makers tend to focus on the most critical habitats rather than promoting comprehensive ecological restoration. Therefore, a scientific framework is needed to identify important habitats for species survival and carry out restoration efforts that consider biodiversity and habitat quality.

Birds are essential to urban ecosystems, as they provide valuable ecosystem services, including pest control, and the restoration of human mental stress, as well as cultural services related to natural perception [8–10]. Furthermore, their sensitivity to changes in habitat makes them reliable indicators of the health of urban ecosystems, as changes in their abundance and diversity can reflect the overall condition of an ecosystem [11]. Rapid landscape changes in urban areas have significant impacts on bird survival. The replacement of urban vegetation with impervious surfaces results in a decrease in the number of bird habitats. Moreover, urban noise, human activities, and light pollution can interfere with bird habitats, leading to a decline in habitat quality [12,13].

Previous studies have mainly considered birds or their habitats from a single perspective. Hotspot areas with a high prevalence of bird species play a crucial role in restoration and conservation efforts to preserve bird diversity [14,15]. For instance, Ruocheng et al. [16] utilized citizen-science data and the Maxent model to assess hotspots of bird distribution and identify conservation gaps in China, thereby contributing to the formulation of habitat protection strategies. Similarly, Arcos et al. [17] identified hotspots of marine bird diversity to assess priority areas for bird conservation. From a habitat perspective, Ganatsas et al. [18] assessed the ecological status of the Salix alba floodplain forests in Kerkini National Park to maintain their support for bird diversity in the region. Neglecting the correlation between bird diversity and habitat quality can lead to a misallocation of resources and impede precise habitat restoration efforts. Only a handful of studies have investigated the complex interdependence between birds and their habitats. Wang et al. [19] evaluated the restoration priority of wetlands by considering the habitat suitability index and the weight of migratory birds. Ferrarini et al. [20] prioritized wetland bird conservation based on bird diversity and threats. Gonzalez et al. [21] identified areas of high bird diversity in the Andes Mountains of Colombia and determined the priority of forest protection based on regional restoration planning. However, factors that reflect habitat quality have not been fully considered, and a more holistic framework that accounts for the relationship between birds and their habitats has yet to be established.

The habitat types in high-density urban areas are more complex and are highly susceptible to human disturbance, making habitat restoration for birds a persistent challenge. Thus, this study proposes a comprehensive framework for prioritizing habitat restoration by considering the matching relationship between bird diversity and habitat quality based on identifying hotspots of habitats suitable for bird survival. This approach enables policy-makers to prioritize the restoration and protection of bird habitats in areas with the highest potential impact, promoting the conservation and sustainable management of urban bird diversity.

Species Distribution Models (SDMs), also known as Ecological Niche Models (ENMs) [22–24], are used to model species distribution based on known distribution points and associated environmental factors. Simulating bird distribution at a large spatial scale requires a substantial quantity of bird data. However, traditional bird community surveys, employing structured methods such as point or transect counts [25,26], entail significant time and manpower costs for collecting bird data. Consequently, these limitations impede large-scale research on bird diversity. The advancement of data collection technology has facilitated the study of bird diversity [27]. Previous studies by Sullivan et al. [28], Squires et al. [29], Liu et al. [30], and Wong et al. [31] have used data from bird observation websites, such as eBird, the China Bird Watching Record Center, iNaturalist, and Burungnesia, to



analyze bird communities and detect bird diversity patterns at different scales. However, there is still limited research utilizing citizen-science data in SDMs [32–34].

Accordingly, this study concentrated on the highly urbanized area of Shanghai, which covers 1278.47 km$^2$ (Figure 1), to carry out the following processes: (1) collect bird data from open-source citizen-science bird observation websites in central Shanghai and use the random forest (RF) model to map bird richness patterns; (2) identify bird hotspot areas in central Shanghai; and (3) accurately delineate habitat restoration priorities with a comprehensive consideration of both bird richness and habitat quality in hotspot areas.

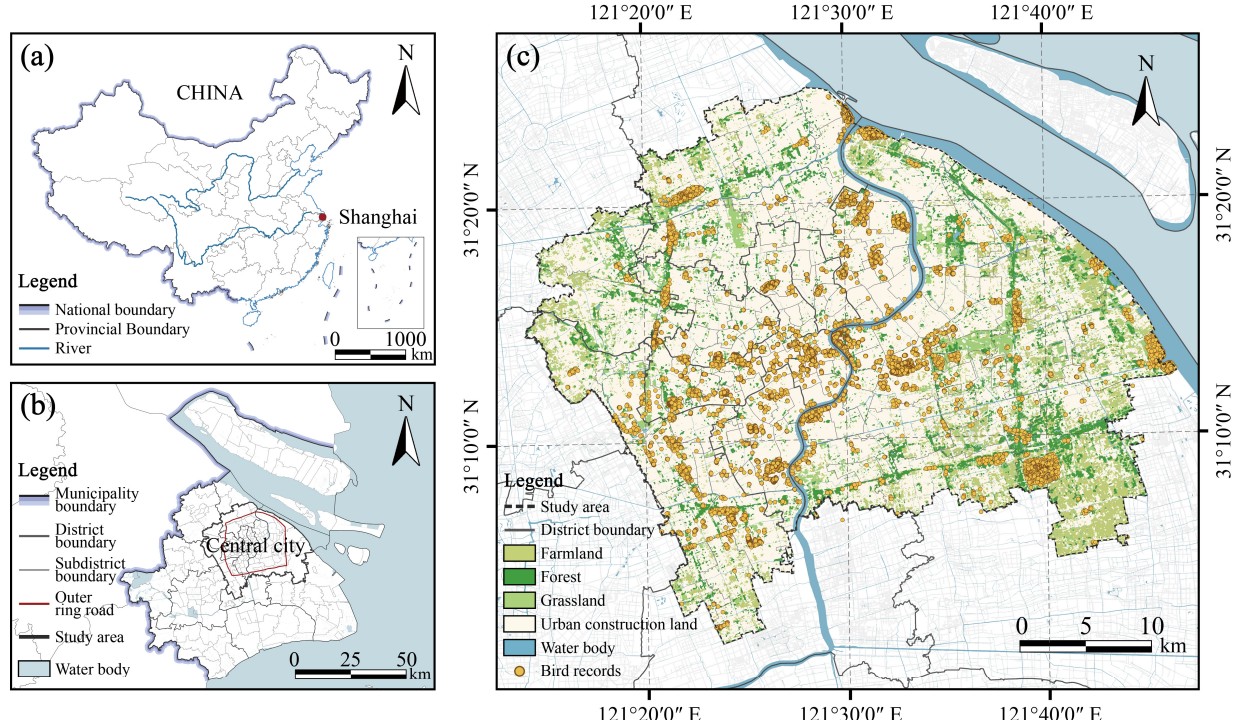

**Figure 1.** Overview of the study area: (**a**) location of Shanghai; (**b**) location of the study area; (**c**) land use types in 2020 and spatial distribution of bird records.

## 2. Study Area and Data Pre-Processing

### 2.1. Study Area

Shanghai is a crucial transit station for migratory birds traveling along the East Asia–Australasia flyway [35]. As of 2019, Shanghai had a total of 494 bird species from 78 families and 22 orders, accounting for 33.51% of all Chinese bird species [36]. However, the rapid expansion of urban areas in Shanghai has led to habitat loss and fragmentation, threatening the city's biodiversity. To support the "14th Five-Year Plan" for ecological space construction and urban environment optimization in Shanghai, the city aims to establish a systematic and efficient pattern of biodiversity conservation spaces by 2035. Therefore, accurately identifying bird hotspots and prioritizing the restoration and protection of bird habitats is essential for scientifically and precisely conserving urban biodiversity.

### 2.2. Bird Records

In this study, a total of 494 potential bird species in Shanghai were selected based on relevant checklists on the classification and distribution of birds of China and Shanghai [36,37]. Bird observation data were collected from https://ebird.org/map (accessed on 21 April 2023) and http://www.birdreport.cn/ (accessed on 21 April 2023) for the period of 2010 to 2023 in central Shanghai. The birding reports provided by the China Bird Watching website, which included observational data from multiple individual reports, were combined to enhance the verification of the observed quantities of bird species and to filter out duplicate observations. From this comprehensive data, a total of 17,461 observation records were selected, representing

311 bird species, which accounted for 62.96% of the total number of species recorded in the bird checklist of Shanghai [37]. Among them, there were four critically endangered species, four endangered species, four vulnerable species, and fifteen species with the near-threatened conservation status (see Table S1 in the Supplementary Materials for the IUCN categories of each species) [36]. In accordance with the classification proposed by Zheng et al. [36], birds were categorized into four groups: resident birds, summer migratory birds, winter migratory birds, and transient birds (Table 1). This classification served as the basis for determining the phenological statuses of bird species in Shanghai. Detailed information regarding the phenological status of each species in our study can be found in Table S1 of the Supplementary Materials.

**Table 1.** Definition of bird phenological statuses [36].

| Phenological Status | Definition |
| --- | --- |
| Resident birds | Birds that reside year-round within their habitat are collectively referred to as resident birds. |
| Summer migratory birds | Migratory birds that breed in a specific region during summer, migrate to warmer southern regions for winter, and return to the same region for breeding the following spring are referred to as summer migratory birds in that particular area. |
| Winter migratory birds | Birds that winter in a specific region, fly north for breeding in the following spring, and return to the same region for wintering in the autumn are referred to as winter migratory birds in that particular area. |
| Transient birds | Birds that pass through a specific area during migration but do not breed or winter in that area are referred to as transient birds in that particular region. |

To assess the quality of citizen-science data, we selected 20 sites during the breeding season in May 2023 and employed a point-count method for bird species data collection [25]. Between 8:00 and 18:00, we conducted bird counts within a 25 m radius for 10 min at each sampling point, ensuring a minimum spacing of 150 m between points [38]. Further details regarding the bird count results can be found in the Supplementary Materials in Tables S2 and S3. To evaluate the consistency between field observations and citizen-science data, we utilized the species repetition rate. This rate represents the proportion of species richness observed both in the field and through citizen science, relative to the total species richness observed across the field and citizen-science data at a specific site. The calculated species repetition rate was $0.78 \pm 0.14$ (Table S4 in the Supplementary Materials), demonstrating the suitability of citizen-science data for SDMs.

*2.3. Environmental Predictors*

To identify the environmental factors affecting bird distribution, we selected factors from five dimensions: climate, topography, landscape pattern, vegetation, and built environment. These factors were chosen based on previous studies [30,39,40] and are listed in Table 2. In order to improve the accuracy of SDMs, we adopted a differentiated approach when selecting environmental factors, taking into account their temporal variability, particularly in relation to Bio1 (precipitation) and Bio2 (temperature). Therefore, we considered the phenological states of bird species when choosing the environmental factors. For resident and transient birds, the annual precipitation and mean temperature were selected as factors. For summer and winter migratory birds, the precipitation and mean temperature in summer and winter were selected as factors, respectively. After resampling using ArcGIS 10.8, each factor was standardized to a spatial scale of 30 m × 30 m. To avoid multicollinearity among the factors, the "corrplot" package in RStudio was used to calculate the correlation between factors (Figure 2) [41]. Bio12 was removed due to its high correlation with Bio1.1, Bio1.2, Bio1.3, Bio2.1, Bio2.2, and Bio2.3. As a result, 11 environmental factors were retained for each category of birds. The results for each factor are shown in Appendix A.

**Table 2.** Environmental factors used in the model.

| Factor Type | Factor Name | | Unit | Source |
|---|---|---|---|---|
| Climate | Bio1 Precipitation | Bio1.1 Annual cumulative precipitation | mm | Calculated using Kriging from spatial analyst tools in ArcGIS 10.8 based on data from https://data.cma.cn/ (accessed on 25 April 2023) |
| | | Bio1.2 Summer cumulative precipitation | mm | |
| | | Bio1.3 Winter cumulative precipitation | mm | |
| | Bio2 Temperature | Bio2.1 Annual mean temperature | °C | Calculated using Kriging from spatial analyst tools in ArcGIS 10.8 based on data from https://data.cma.cn/ (accessed on 25 April 2023) |
| | | Bio2.2 Summer mean temperature | °C | |
| | | Bio2.3 Winter mean temperature | °C | |
| Topography | Bio3 Elevation | | m | https://www.gscloud.cn/ (accessed on 23 April 2023) |
| | Bio4 Slope | | ° | Calculated using slope from spatial analyst tools in ArcGIS 10.8 based on DEM data |
| Landscape pattern | Bio5 Landscape Shannon's Diversity Index (SHDI) | | - | Calculated using Fragstats 4.2 based on land use data |
| Vegetation | Bio6 Normalized Difference Vegetation Index (NDVI) | | - | Calculated using ENVI 5.4 based on Landsat 8 OLI TIRS data from https://www.gscloud.cn/ (accessed on 23 April 2023) |
| Built environment | Bio7 Land use | | - | Calculated using ENVI 5.4 based on Landsat 8 OLI TIRS data from https://www.gscloud.cn/ (accessed on 23 April 2023) |
| | Bio8 Building density | | - | Calculated using KernelDensity from spatial analyst tools in ArcGIS 10.8 based on building data |
| | Bio9 Building height | | m | From the literature [42] |
| | Bio10 Distance to water body | | m | Calculated using MultipleRingBuffer from analysis tools in ArcGIS 10.8 based on water data |
| | Bio11 Distance to roads | | m | Calculated using MultipleRingBuffer from analysis tools in ArcGIS 10.8 based on road data |
| | Bio12 Artificial night-time light | | - | https://poles.tpdc.ac.cn/zh-hans/data/e755f1ba-9cd1-4e43-98ca-cd081b5a0b3e/ (accessed on 24 April 2023) |

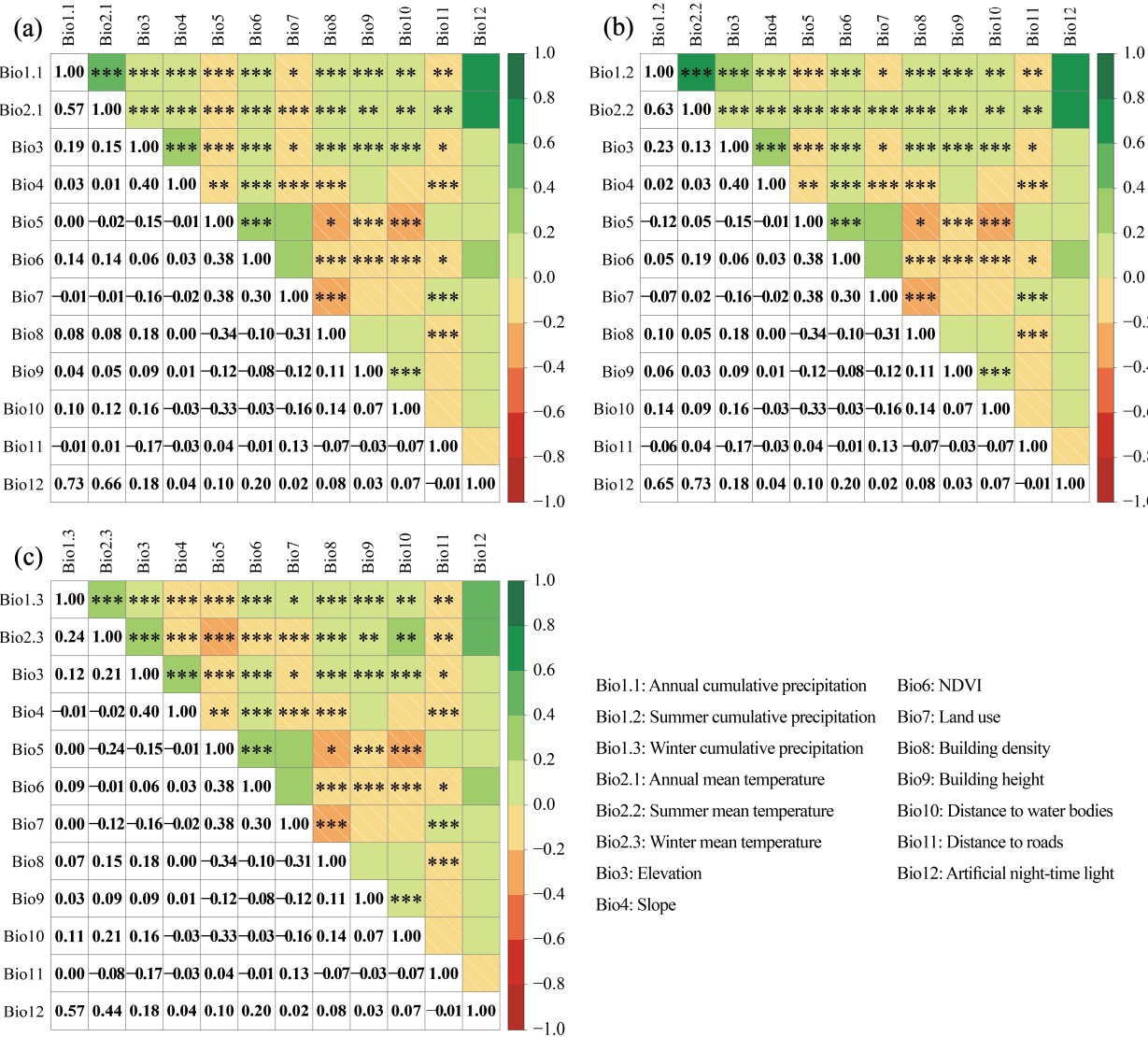

**Figure 2.** Correlation between environmental factors for (**a**) resident and transient/(**b**) summer migratory/(**c**) winter migratory birds. *, **, and *** represent statistical significance at $p < 0.05$, $p < 0.01$, and $p < 0.001$ levels, respectively.

## 3. Methods

### 3.1. Research Framework

The methodological framework of this study is depicted in Figure 3. We used the RF model to predict the distribution of bird species and overlaid the distribution of each species to obtain the spatial pattern of bird richness across the region. Local spatial autocorrelation indices were then computed to identify hotspots of bird richness. To assess the quality of bird habitats, we used the InVEST model [43]. Finally, the bird richness of the hotspot area was matched with the habitat quality to determine the priority of bird habitat restoration.

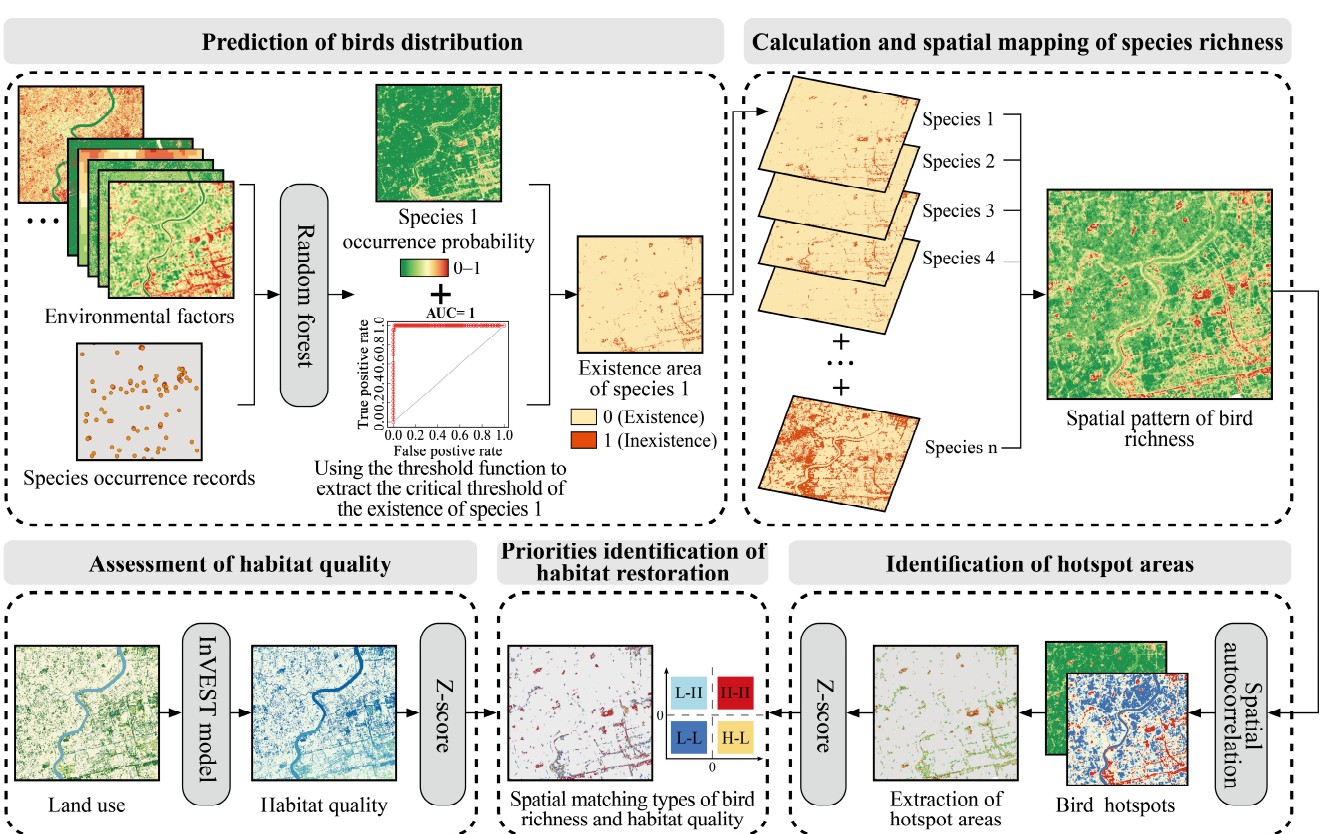

**Figure 3.** Research framework.

### 3.2. Prediction of Bird Distribution

Machine learning algorithms, such as Maximum Entropy (Maxent), RF, the Generalized Linear Model (GLM), and Classification and Regression Tree (CART), have demonstrated better performance in species prediction than traditional regression algorithms [44]. Although the Maxent model is commonly used for species prediction simulation [16,45–47], studies have shown that the RF model outperforms it in predicting species distribution and can provide more accurate simulation results with fewer species records [39,40]. While the RF model has mainly been used for simulating single species, its potential for identifying bird diversity hotspots remains underutilized.

RF is a machine learning algorithm that combines multiple decision trees using bootstrap sampling. It has high accuracy in data classification, regression, and prediction. In this study, we used the "randomForest" package in RStudio to simulate species distribution [48] with the model parameter Ntree set to 500. To determine the accuracy of our model fitting results, we used the Area Under the Curve (AUC) of the Operating Characteristic Curve (ROC), where AUC values closer to 1 indicate higher prediction accuracy. Furthermore, an RF regression analysis can quantify the importance of environmental factors in shaping bird distribution patterns using the IncNodePurity value [39].

To ensure the accuracy and reliability of our RF model, we used a sample size referenced from previous research [40] and excluded bird species with less than 6 observation records. A total of 198 simulated species were selected, including 49 resident birds, 32 summer migratory birds, 66 winter migratory birds, and 51 transient birds. For each individual species, the threshold function was used to calculate the optimal threshold for species presence and thus obtain the binary distribution of species presence. To obtain the distribution patterns of bird richness, the binary results of each species were overlaid using the following formula [30]:

$$Richness = s_1 + s_2 + s_3 + \ldots + s_i \tag{1}$$

where *Richness* is the sum of the total number of bird species.

### 3.3. Identification of Hotspot and Coldspot Areas

Hotspot analysis (local Moran's *I*) is an effective model for identifying the spatial distribution of spatial clustering patterns. We used the GeoDa software to calculate the local Moran's *I* and identify hotspots and coldspots [49], which helped us detect the spatial clustering degree of bird richness values and their surrounding values. The formula for the local Moran's *I* is as follows [50]:

$$I_i = \frac{z_i - \bar{z}}{\sigma^2} \sum_{j=1,j\neq i}^{n} \left[ w_{ij}(z_j - \bar{z}) \right] \tag{2}$$

where $z_i$ represents the value of the spatial variable $z$ at location $i$. $\bar{z}$ is the mean value of variable $z$. $z_j$ represents all the values of variable $z$ outside the $i$ location. $\sigma^2$ is the variance of variable $z$. $w_{ij}$ is the spatial weight.

### 3.4. Assessment of Habitat Quality

The InVEST model is widely used for assessing habitat quality [7,18,51]. In this study, we used InVEST 3.7.0 to evaluate habitat quality in central Shanghai. The habitat quality was obtained by assessing habitat suitability and degradation with land use data and their sensitivity to various threat sources, as well as the impact distances of threat factors. The formulas are articulated as follows [18]:

$$D_{xj} = \sum_{r=1}^{R} \sum_{y=1}^{Y_r} (w_r / \sum_{r=1}^{R} w_r) r_y i_{rxy} \beta_x S_{jr} \tag{3}$$

$$i_{rxy} = 1 - \left( \frac{d_{xy}}{d_{rmax}} \right) if \ linear \tag{4}$$

$$i_{rxy} = exp(-(\frac{2.99}{d_{rmax}}) \times d_{xy}) \ if \ exponential \tag{5}$$

where $D_{xj}$ denotes the habitat degradation level of raster $x$ in land use type $j$. $R$ is the number of threat factors. $Y_r$ is the total number of raster grids for threat factor $r$. $w_r$ is the weight of threat factor $r$. $r_y$ is the number of threat factors for each raster grid of land use type. $i_{rxy}$ represents the impact of threat factor $r$ on each habitat raster grid, which can be calculated using linear or exponential functions (Equations (4) and (5)). $\beta_x$ is the degree of protection for the raster grid. $S_{jr}$ is the sensitivity of land use type $j$ to threat factor $r$, ranging from 0 to 1. $d_{xy}$ is the distance between the land use raster and the threat factor. $d_{rmax}$ is the influence range of threat factor $r$.

$$Q_{xj} = H_j(1 - (\frac{D_{xj}^z}{D_{xj}^z + k^z})) \tag{6}$$

where $Q_{xj}$ represents the habitat quality. $H_j$ denotes the ecological suitability of land use type $j$. $k$ is the half-saturation coefficient. $z$ is the default parameter of the model.

As shown in Tables 3 and 4, the various parameters in the above formula were referenced from previous studies conducted in highly urbanized areas with similar natural characteristics, population sizes, and economic levels to Shanghai [7,51,52].

**Table 3.** Threat factor parameter settings.

| Threat | Max Distance (km) | Weight | Decay |
|---|---|---|---|
| Farmland | 1.00 | 0.30 | Exponential |
| Urban construction land | 12.0 | 1.00 | Exponential |
| Building | 7.00 | 0.80 | Linear |
| Road | 5.00 | 0.60 | Linear |

**Table 4.** Sensitivity of land use types to each threat.

| Land Use | Habitat | Threat Factors | | | |
|---|---|---|---|---|---|
| | | Farmland | Urban Construction Land | Building | Road |
| Forest | 1.30 | 0.30 | 0.70 | 0.80 | 0.60 |
| Grassland | 0.50 | 0.20 | 0.70 | 0.65 | 0.30 |
| Water body | 1.00 | 0.30 | 0.80 | 0.85 | 0.40 |
| Urban construction land | 0.00 | 0.00 | 0.00 | 0.00 | 0.00 |
| Farmland | 0.30 | 0.00 | 0.60 | 0.70 | 0.20 |

### 3.5. Priority Identification of Habitat Restoration for Birds

To determine the relative relationship between bird richness and habitat quality, we standardized the two variables using *z*-scores, with 0 as the standard. This analysis allowed us to gain a more comprehensive understanding of the relationship between these two variables [53]. The calculation formula for the *z*-score is as follows [54]:

$$z = \frac{x_i - \overline{x}}{s} \tag{7}$$

where *z* represents the standardized value of the spatial unit. $x_i$ is the value of bird richness or habitat quality for the spatial unit *i* in the hotspot areas. $\overline{x}$ is the average bird richness or habitat quality. *s* is the variance of bird richness or habitat quality.

## 4. Results

### 4.1. Potential Species Richness Patterns and the Importance of Environmental Factors

As shown in Figure 4a, the richness of all bird species observed in central Shanghai ranged from 0 to 150 (details of the simulated distribution of individual species can be found in Appendix B) and exhibited a significant spatial clustering pattern (Moran's $I$ = 0.870, $p$ = 0.001, $z$ > 2.58). High values of bird richness were concentrated in the outer-ring structural green belt and urban comprehensive parks, as well as green spaces along rivers and coastal areas. When considering the four categories of bird residency, high values of resident bird richness were found to have a wider spatial distribution compared to the other three categories (Figure 4b), with a maximum value of 45. High values were concentrated in various large urban parks, such as Gongqing Forest Park in the riverside area, Pudong Riverside Forest Park near the estuary, Wusong Battery Wetland Park, and Century Park, in high-density urban areas. The diversity patterns of summer migratory birds, winter migratory birds, and transient birds were consistent (Figure 4c–e), with maximum values of 26, 56, and 34, respectively. High-diversity areas of these three categories were mostly located in large urban parks, such as Gongqing Forest Park, Xing Jiangwan City Park, Century Park, Pudong Riverside Forest Park, and Wusong Battery Wetland Park.

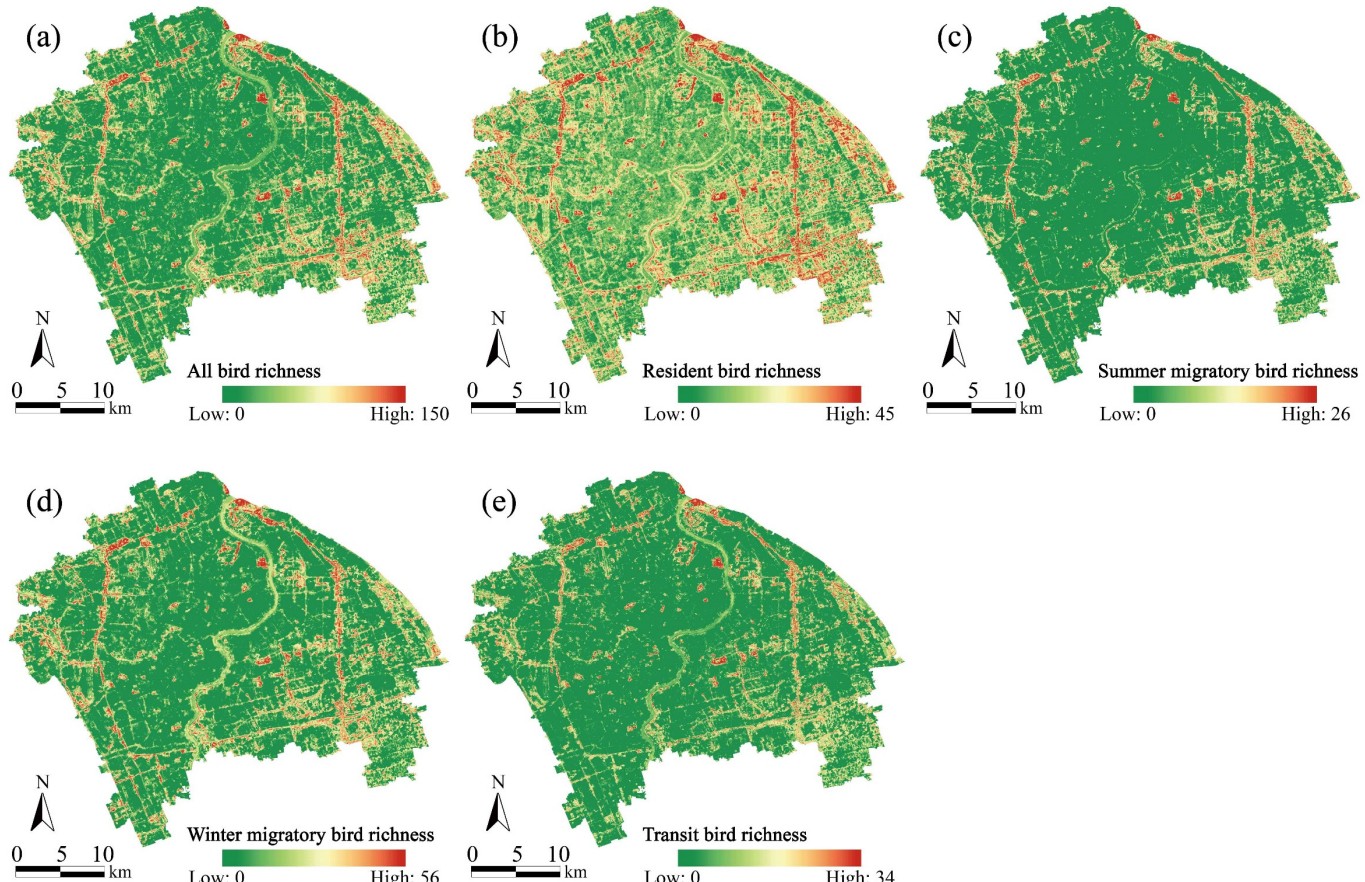

**Figure 4.** Spatial pattern of bird richness. (**a–e**): All/resident/summer migratory/winter migratory/transient bird richness.

The influence of environmental factors on different bird species was assessed using the RF model, and IncNodePurity values were calculated. However, due to variations in the ranges of IncNodePurity values among species, we normalized the importance of each factor. This allowed us to gain insights into the underlying mechanisms driving bird species distribution (Figure 5).

Among the analyzed factors, Bio6 (NDVI) exhibited the highest median importance for resident birds (0.90), winter migratory birds (0.78), and transient birds (0.85). It ranked second for summer migratory birds (0.83). Bio5 (Landscape SHDI) was also identified as a significant driving factor for bird distribution in central Shanghai. It held the highest rank for summer migratory birds (1.00) and the second rank for resident birds (0.66), winter migratory birds (0.68), and transient birds (0.70). Bio8 (building density) secured the third rank for resident birds (0.62), the second rank for summer migratory birds (0.56), the third rank for winter migratory birds (0.54), and the fifth rank for transient birds (0.41). Conversely, Bio9 demonstrated the least importance across all bird categories, with a median value of 0 for each category.

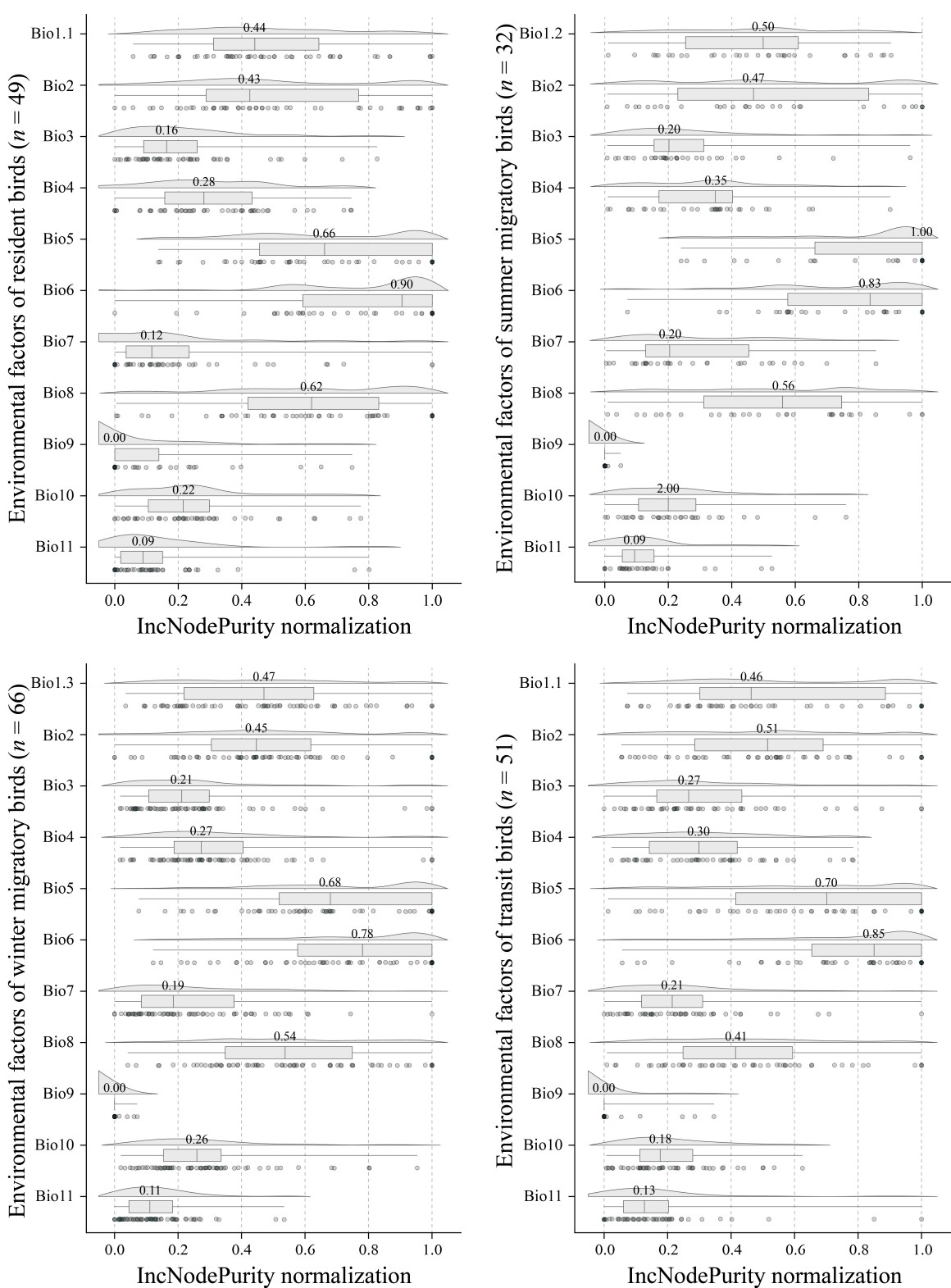

**Figure 5.** Importance of environmental factors for each type of bird. The points in the figure represent the normalized IncNodePurity values of each species for every environmental factor.

### 4.2. Spatial Pattern of Bird Hotspots

Figure 6 displays statistically significant bird hotspots and coldspots (with a confidence level higher than 95, i.e., $p < 0.05$) in central Shanghai. The hotspots are mainly concentrated in the outer-ring structural green belt, large park green spaces, and ecological spaces along

rivers and coasts, covering a total area of 248.41 km$^2$. Forest is the dominant land use type within the hotspots, accounting for 45.23% of the hotspot areas (112.31 km$^2$). Although forests only comprise 17.33% of the study area (221.61 km$^2$), almost half of them are bird hotspots, suggesting the importance of forests for urban birds. Apart from forest, the land use composition of the hotspots, ranked from high to low, is urban construction land (52.36 km$^2$, accounting for 21.09% of the hotspot areas), followed by grassland (47.48 km$^2$, accounting for 19.12%), water bodies (22.17 km$^2$, accounting for 8.93%), and farmland (13.97 km$^2$, accounting for 5.63%).

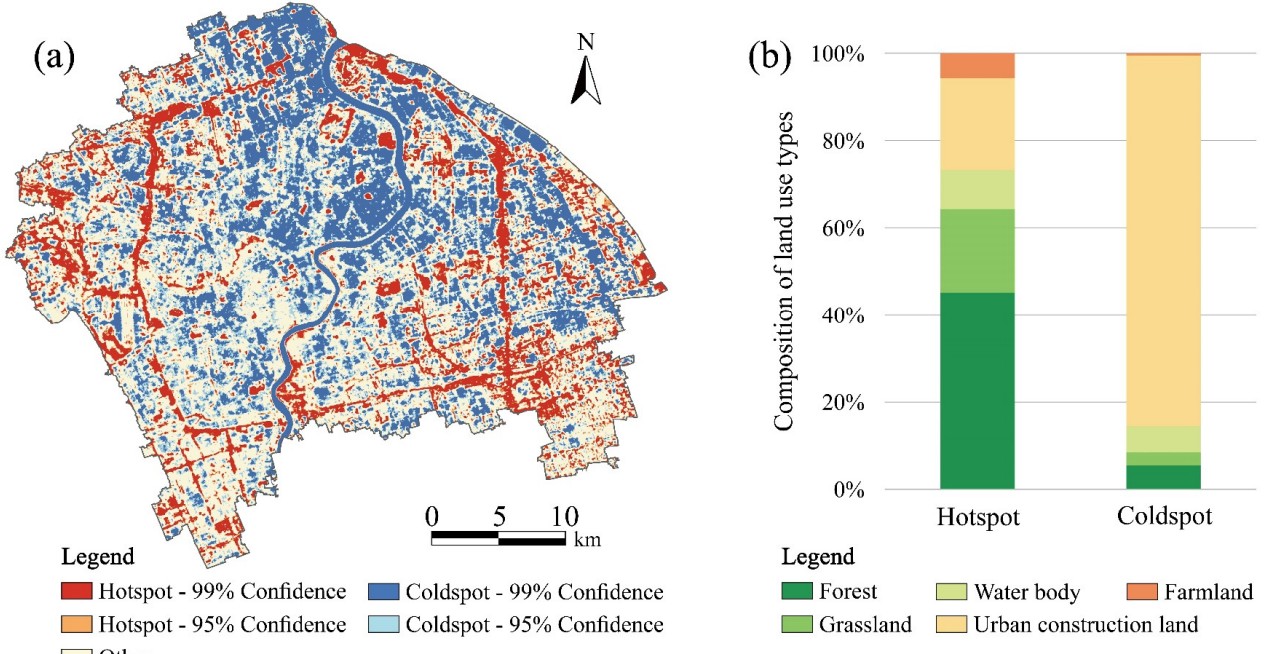

**Figure 6.** Identification of bird hotspots. (**a**) The spatial distribution of hotspots and coldspots based on local Moran's *I*; (**b**) the proportion of each land use type in hotspot/coldspot areas.

The coldspots for birds cover an area of 470.85 km$^2$, mainly concentrated in the urban construction land of the central area, with an area of 400.55 km$^2$, accounting for 85.07% of the coldspot areas. Water bodies, forest, and grassland are the next most common habitat types in the coldspots, with areas of 27.48 km$^2$ (accounting for 5.84%), 25.65 km$^2$ (accounting for 5.45%), and 14.65 km$^2$ (accounting for 3.11%), respectively. Farmland has the lowest land cover, covering only 2.52 km$^2$, accounting for 0.53% of the coldspot areas.

### 4.3. Evaluation and Spatial Distribution of Habitat Quality

The habitat quality in central Shanghai varies from 0.00 to 0.74 (Figure 7), with significant spatial variations in distribution patterns. High-quality habitat areas (V) consist mainly of forest and grassland, covering 133.73 km$^2$ downstream of the Huangpu River, in the outer-ring green belt, and large urban green spaces. In contrast, low-quality habitat areas (I) are primarily urban construction land in the city center, covering 811.83 km$^2$, and are heavily impacted by construction activities.

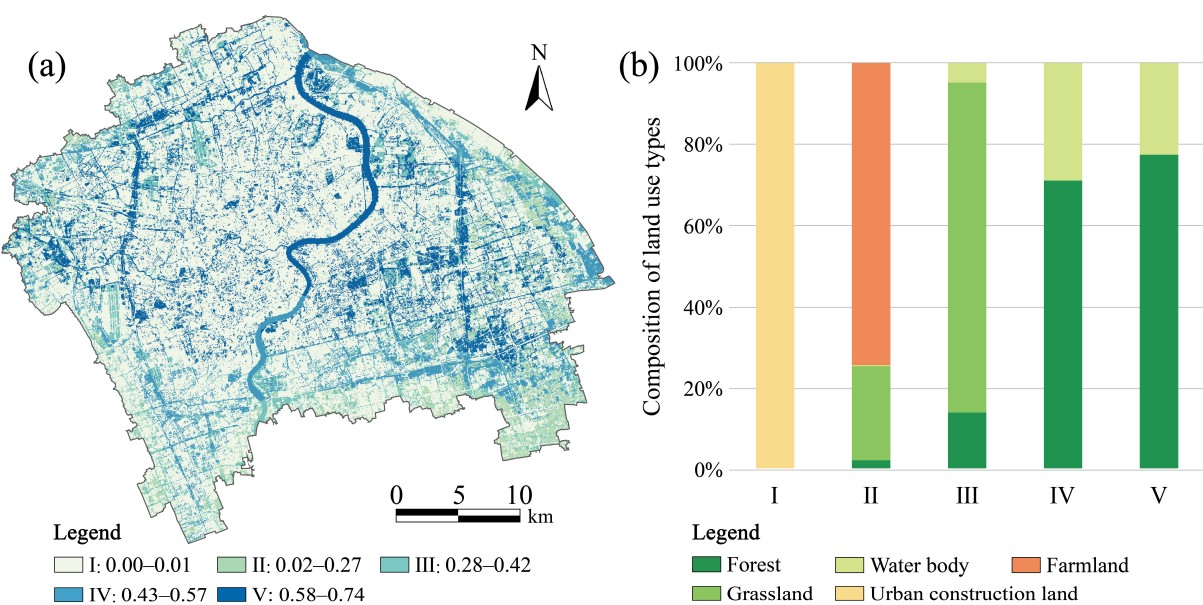

**Figure 7.** (**a**) Spatial distribution of habitat quality; (**b**) the proportion of land use for each level of habitat quality.

### 4.4. Spatial Matching Types of Bird Richness and Habitat Quality

The matching relationship between bird richness and habitat quality, standardized by *z*-score, is illustrated in Figures 8 and 9 and Table 5. Based on the matching results, four types were identified, and their spatial areas ranked from high to low are: "Low bird richness-Low habitat quality (LBR-LHQ)" (81.80 km², accounting for 32.95% of the hotspot area), "High bird richness-High habitat quality (HBR-HHQ)" (73.25 km², accounting for 29.50%), "Low bird richness-High habitat quality (LBR-HHQ)" (58.56 km², accounting for 23.58%), and "High bird richness-Low habitat quality (HBR-LHQ)" (34.68 km², accounting for 13.97%).

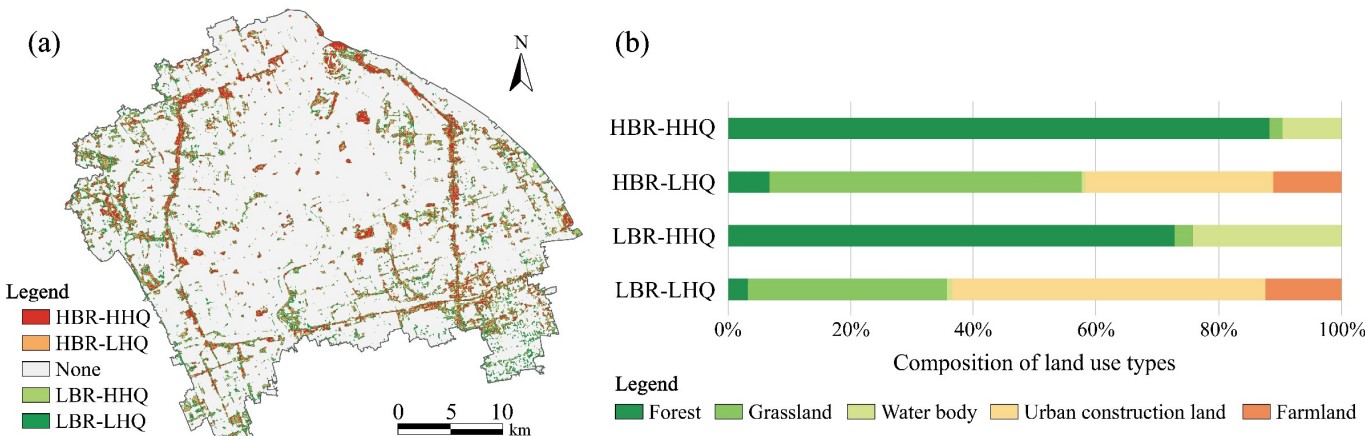

**Figure 8.** (**a**) Spatial matching types of bird richness and habitat quality in hotspot areas; (**b**) the proportion of land use for each type of restoration.

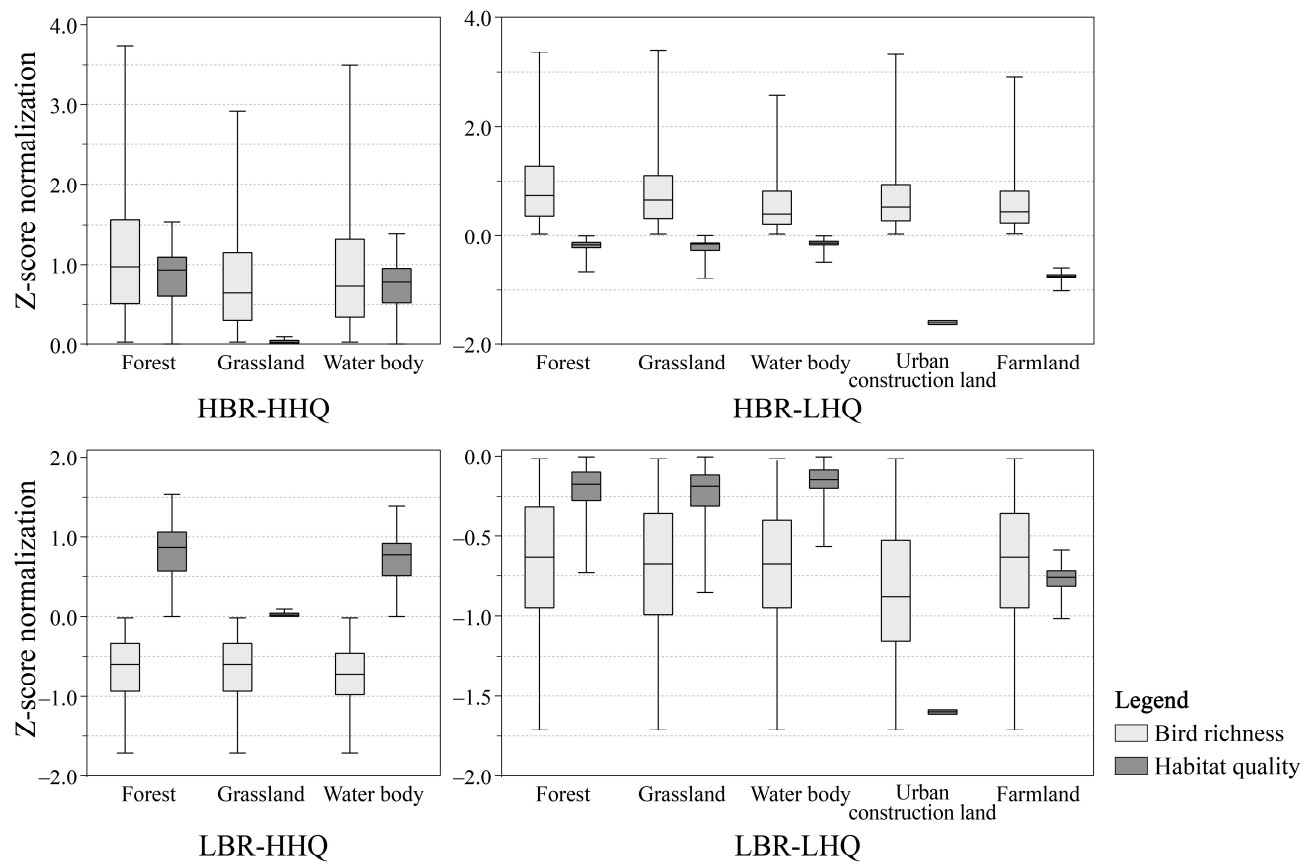

**Figure 9.** Four classifications of standardized bird richness and habitat quality.

**Table 5.** The land use area of the four restoration types and the mean values of two variables.

| Spatial Type | Land Use Area (km$^2$) | | | | | | Average of Bird Richness | Average of Habitat Quality |
| | Forest | Grassland | Water Body | Urban Construction Land | Farmland | Total | | |
|---|---|---|---|---|---|---|---|---|
| HBR-HHQ | 64.66 | 1.55 | 7.04 | 0.00 | 0.00 | 73.25 | 94.74 (1.05 *) | 0.57 (0.83 *) |
| HBR-LHQ | 2.33 | 17.67 | 0.22 | 10.60 | 3.85 | 34.68 | 86.90 (0.72 *) | 0.21 (−0.69 *) |
| LBR-HHQ | 42.67 | 1.69 | 14.19 | 0.00 | 0.00 | 58.56 | 53.25 (−0.67 *) | 0.56 (0.77 *) |
| LBR-LHQ | 2.64 | 26.56 | 0.72 | 41.76 | 10.12 | 81.80 | 50.82 (−0.77 *) | 0.14 (−0.99 *) |

* represents the mean value of standardized bird richness/habitat quality.

The LBR-LHQ spatial type is predominantly distributed along roads and rivers, mostly consisting of urban construction land covering an area of 41.76 km$^2$ (51.05% of this type). Grassland is the second most common land cover type, covering an area of 26.56 km$^2$ (32.47%). Farmland, forest, and water bodies are relatively uncommon, covering areas of 10.12 km$^2$ (12.37%), 2.64 km$^2$ (3.23%), and 0.72 km$^2$ (0.88%) of this type, respectively. The standardized relationship between bird richness and habitat quality (Figure 9) shows distinct patterns. Forest and farmland display the highest bird richness in this type, with a median value of −0.64. Grassland and water bodies follow closely with a median value of −0.68. Conversely, urban construction land exhibits the lowest bird richness, scoring a median value of −0.88. In terms of habitat quality, water bodies rank highest within this type, boasting a median value of −0.15, followed by forest (−0.18), grassland (−0.19), and farmland (−0.76). Urban construction land, on the other hand, presents the lowest habitat quality, with a median value of −1.60. The HBR-HHQ spatial type is primarily characterized by forest, which is distributed in the outer-ring green belt and large urban parks, covering an area of 64.66 km$^2$ and accounting for 88.27% of this type. Water bodies

and grassland are the second most common land cover types, covering areas of 7.04 km$^2$ and 1.55 km$^2$ and accounting for 9.61% and 2.12% of the HBR-HHQ spatial type, respectively. According to Figure 9, the results suggest that forest exhibits the highest levels of bird richness and habitat quality within this type, with median values of 0.98 and 0.94, respectively. Water bodies rank second, with median values of 0.73 for bird richness and 0.78 for habitat quality. In contrast, grassland exhibits the lowest bird richness and comparatively lower habitat quality, with median values of 0.65 and 0.02, respectively.

The LBR-HHQ spatial type is primarily distributed near roads, water systems, and some agricultural and forestry spaces. Forest is the dominant land cover type in this spatial pattern, covering an area of 42.67 km$^2$ and accounting for 72.87% of this type. Water bodies and grassland are the second most common land cover types, covering areas of 14.19 km$^2$ (24.24%) and 1.69 km$^2$ (2.89%), respectively. Figure 9 illustrates that within this type, forest and grassland exhibit the highest bird richness, with a median value of −0.59. In contrast, water bodies demonstrate relatively lower bird richness, with a median value of −0.72. Additionally, forest stands out as having the highest habitat quality (0.87), followed by water bodies (0.77), while grassland shows comparatively lower habitat quality, with a median value of 0.02.

The HBR-LHQ spatial type has the lowest proportion of the total area and is mainly nested between large ecological patches. This spatial pattern consists primarily of grassland and urban construction land, covering areas of 17.67 km$^2$ and 10.60 km$^2$, accounting for 50.96% and 30.58% of this type, respectively. Farmland, forest, and water bodies are relatively less common, covering areas of 3.85 km$^2$, 2.33 km$^2$, and 0.22 km$^2$ and accounting for 11.11%, 6.73%, and 0.63% of the total area, respectively. Based on Figure 9, the results indicate that forest has the highest bird richness within this type, with a median value of 0.73. It is closely followed by grassland (0.65), urban construction land (0.52), farmland (0.44), and water bodies (0.40). Regarding habitat quality, water bodies exhibit relatively higher quality, with a median value of −0.14, followed by grassland (−0.15), forest (−0.17), and farmland (−0.74). Notably, urban construction land displays the lowest habitat quality, with a median value of −1.60.

## 5. Discussion

### 5.1. Environmental Drivers of Bird Distribution

Bio6 (NDVI) has exhibited strong performance in simulating species distribution [55,56]. As the NDVI is closely associated with plant growth, health, and productivity, it provides crucial insights into bird habitats. We recommend that city managers prioritize vegetation improvement to enhance the NDVI, as it holds significant implications for bird diversity conservation. Similarly, Bio5 (Landscape SHDI) plays a pivotal role in shaping bird abundance patterns, which aligns with previous studies by Liu et al. [30] and Matthies et al. [57]. Heterogeneous patches offer diverse habitats for birds, as supported by the findings of Callaghan et al. [58], who reported that urban green spaces with higher heterogeneity host greater bird richness compared to natural green areas. This indicates that green spaces with high diversity can provide a range of habitats and food sources, thereby contributing to the maintenance of a higher level of bird diversity.

The urban landscape, characterized by intensive urbanization and a lack of food sources, poses significant challenges for bird adaptation. Previous studies have demonstrated the adverse effects of Bio8 (building density) and Bio9 (building height) in urban environments on bird richness [59]. Our study further emphasizes the importance of Bio8 over Bio9 in shaping bird richness patterns.

### 5.2. Implications for Bird Habitat Restoration

The HBR-LHQ area, primarily consisting of grassland and urban construction land, is a vital habitat in need of restoration. The coexistence of natural and built environments in this area contributes to its low habitat quality and high bird richness. This coexistence facilitates diverse habitat types and food sources, benefiting urban adapters and urban exploiters,

ultimately leading to an increase in bird richness [60]. However, human disturbances pose a significant threat to this habitat type, resulting in reduced habitat quality. Consequently, restoring the regional habitat is imperative to enhance bird habitat suitability. To achieve this, it is recommended to prioritize the preservation of semi-natural vegetation and maintain the quality of existing forest. Simultaneously, afforestation efforts should be directed towards unused and abandoned lands to increase vegetation coverage and provide additional bird habitats.

The LBR-LHQ area, encompassing more than half of the urban construction land, presents a challenge for bird habitation due to relatively low habitat quality despite high bird richness in the forest, grassland, and water body habitats. The fragmented nature of these ecological patches necessitates the establishment of ecological corridors to connect them and create a cohesive network of bird habitats within the region. Moreover, optimizing the sizes of forest and grassland areas in the urban landscape and carefully considering their proximity to water sources and other landscape elements are crucial steps. These measures are essential for enhancing the suitability of bird habitats, increasing their chances of survival in urban areas, and ultimately augmenting bird richness.

The LBR-HHQ area primarily consists of forest and water bodies. While the water bodies exhibit relatively high habitat quality, they lack suitable nesting sites for birds, resulting in lower bird richness. On the other hand, the forest exhibits higher habitat quality, albeit still lower compared to the HBQ-HHQ region. To enhance bird richness in this region, a transformation into a composite wetland habitat can be achieved by surrounding the core water body with forest, grassland, and reeds. This integration and connectivity of diverse landscape elements within the habitat will provide birds with varied foraging and nesting conditions, thereby augmenting bird richness.

The HBR-HHQ area represents a well-coordinated ecosystem predominantly characterized by high-quality forest and grassland habitats. Notably, the forest and water bodies within this region exhibit high bird richness. As there is no urban construction land or farmland in this area, it presents favorable conditions for bird survival, emphasizing the need for habitat conservation strategies. City managers should prioritize the protection of forest and bird habitats along riverbanks. Optimizing vegetation community structure, enhancing vegetation diversity, and increasing spatial complexity are recommended measures to sustain the area's high bird richness status.

*5.3. Research Limitations*

Despite proposing a new perspective that integrates bird richness and habitat quality to prioritize habitat restoration in hotspot areas, this study also has certain limitations. Firstly, data collection for biodiversity research inevitably faces the issue of uneven sampling [15]. The utilization of citizen-science data has expanded the spatiotemporal scale of bird pattern analysis, but it also suffers from the issue of observer subjectivity. Observers tend to exhibit a preference for observing and recording birds in green spaces, even though birds are primarily distributed in urban green spaces [61]. This leads to fewer observations and records of birds in urban construction land, resulting in a certain deviation between the spatial distribution data of bird observations and the actual distribution of birds in the highly urbanized central area of Shanghai. Secondly, the conservation and restoration of bird habitats should prioritize the distribution and habitat preferences of rare and endangered bird species. This study utilized bird richness and habitat quality for spatial matching and restoration priority classification but did not further incorporate the endangered statuses of different bird species into the delineation of habitat restoration priority and strategy development, nor did it take into account the interspecific relationships among bird species. Furthermore, the relationship between species and scale selection can also affect the accuracy of model results. Synurbic species such as *Passer montanus*, *Pycnonotus sinensis*, and *Turdus mandarinus* are not sensitive to habitat selection, and fine-grained environmental factors can improve the accuracy of prediction results. However, species with larger ranges that are less common in urban areas, such as *Accipiter trivirgatus*, have

relatively poor prediction results. Their observations in urban areas may be due to flying through the city or foraging in the city, rather than nesting in the city, which can also affect the accuracy of model prediction.

## 6. Conclusions

Our study offers a new perspective for identifying priority areas for bird habitat restoration by integrating bird hotspots and habitat quality. Using the RF model, we mapped the spatial distribution pattern of bird richness and found that large structural green spaces, coastal areas, and riparian green spaces had relatively high bird richness. We calculated local spatial autocorrelation indices to identify hotspot areas of bird richness, covering a total area of 248.29 km$^2$, mainly composed of forested areas accounting for 45.23% of the total area. Urban forests, which occupy 17.33% of the total land area in central Shanghai, contain half of the forested areas identified as bird hotspots, indicating that forest is an essential habitat type for urban birds. Finally, we conducted a matching analysis of the standardized bird richness and habitat quality, dividing the priority of bird habitat restoration into four types: HBR-LHQ, LBR-LHQ, LBR-HHQ, and HBR-HHQ, in descending order. HBR-LHQ is the top priority area that deserves habitat restoration, while HBR-HHQ is the area with high bird richness and good habitat quality that deserves protection. The framework developed in this study can precisely identify the restoration priority of bird habitats and provide an effective tool for urban habitat restoration and biodiversity conservation.

**Supplementary Materials:** The following supporting information can be downloaded at https://www.mdpi.com/article/10.3390/f14081689/s1; Table S1. Bird data collected from public citizen-science websites; Table S2. Area and location information of 20 survey sites; Table S3. Bird data collected from field observations; Table S4: Comparison between observational data and citizen-science data.

**Author Contributions:** Conceptualization, Y.W. and X.L.; methodology, Y.W., X.L. and R.W.; software, X.L. and R.W.; validation, Y.W., X.L., R.W., Y.J. and J.H.; formal analysis, Y.W. and X.L.; investigation, X.L. and R.W.; resources, X.L.; data curation, Y.J.; writing—original draft preparation, Y.W. and X.L.; writing—review and editing, Y.W. and X.L.; visualization, X.L.; supervision, Y.W.; project administration, Y.W.; funding acquisition, Y.W. All authors have read and agreed to the published version of the manuscript.

**Funding:** This work was supported by the National Natural Science Foundation of China (grant No. 52238003).

**Data Availability Statement:** The data are contained within the article, and the details are shown in Table 2.

**Acknowledgments:** We thank Cunyu Xu for her valuable contributions in collecting the data.

**Conflicts of Interest:** The authors declare no conflict of interest. The funders had no role in the design of the study; in the collection, analyses, or interpretation of data; in the writing of the manuscript; or in the decision to publish the results.

# Appendix A. Environmental Factors Used in RF Model

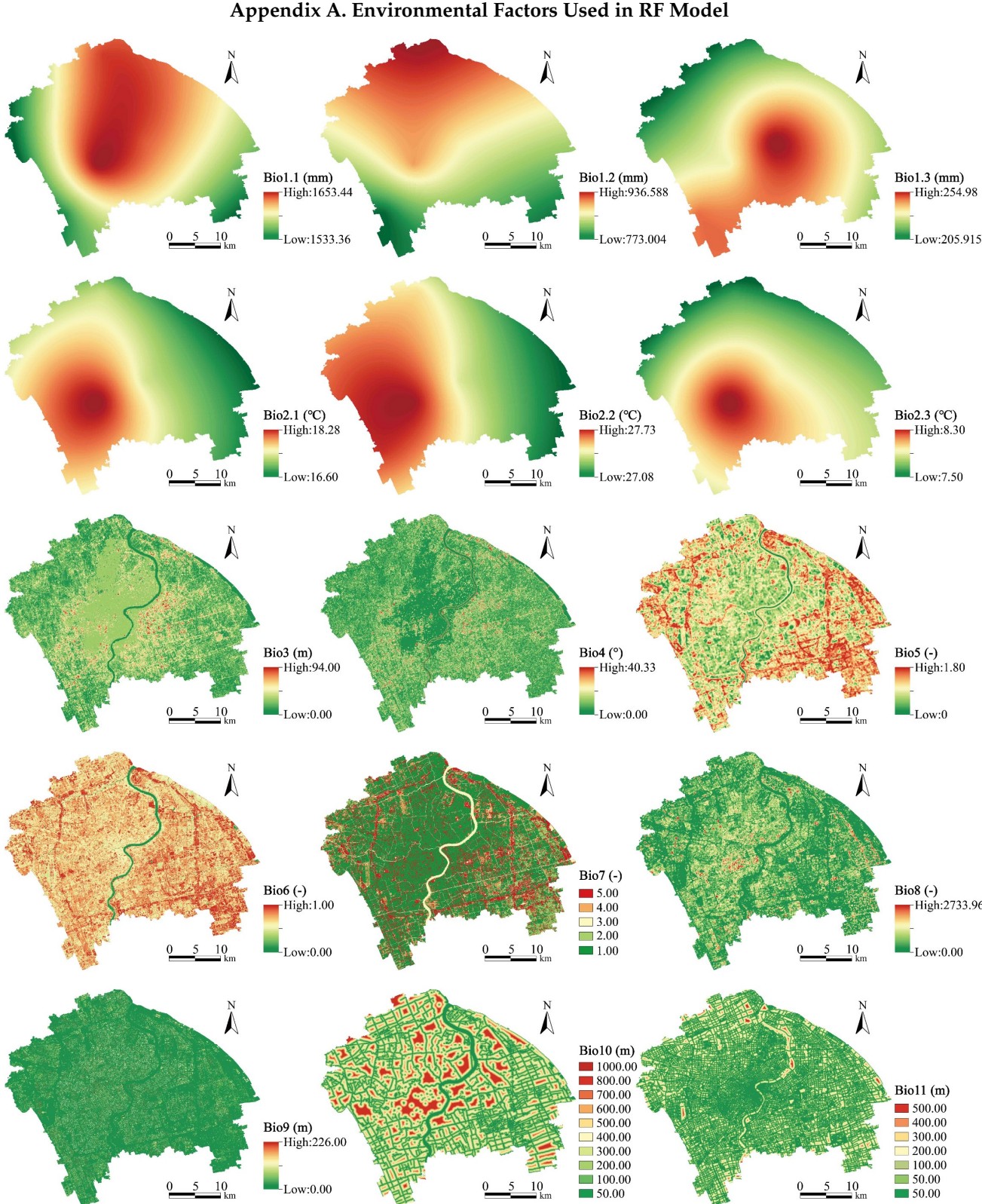

## Appendix B. The Spatial Distribution of the Probability of Occurrence for Individual Species

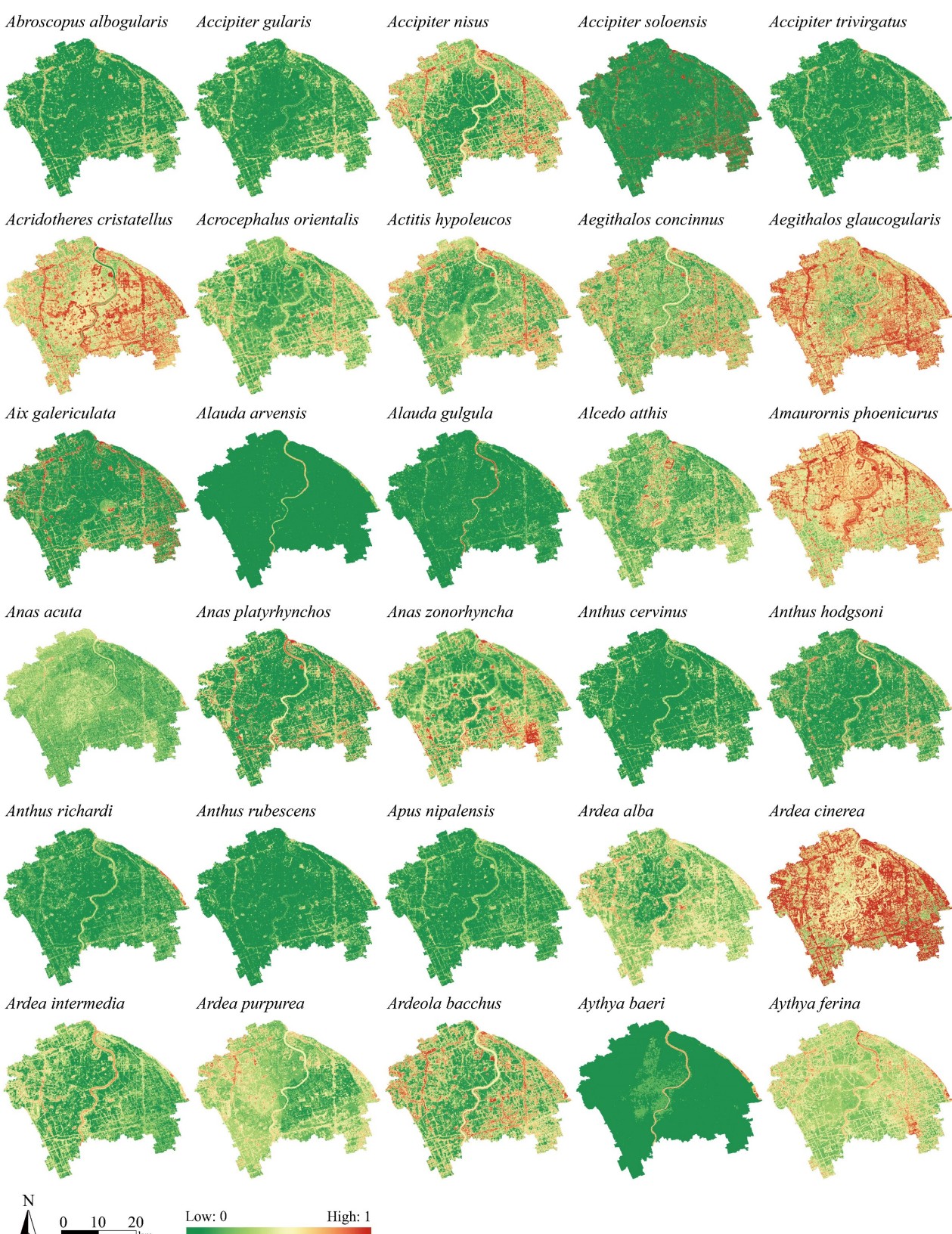

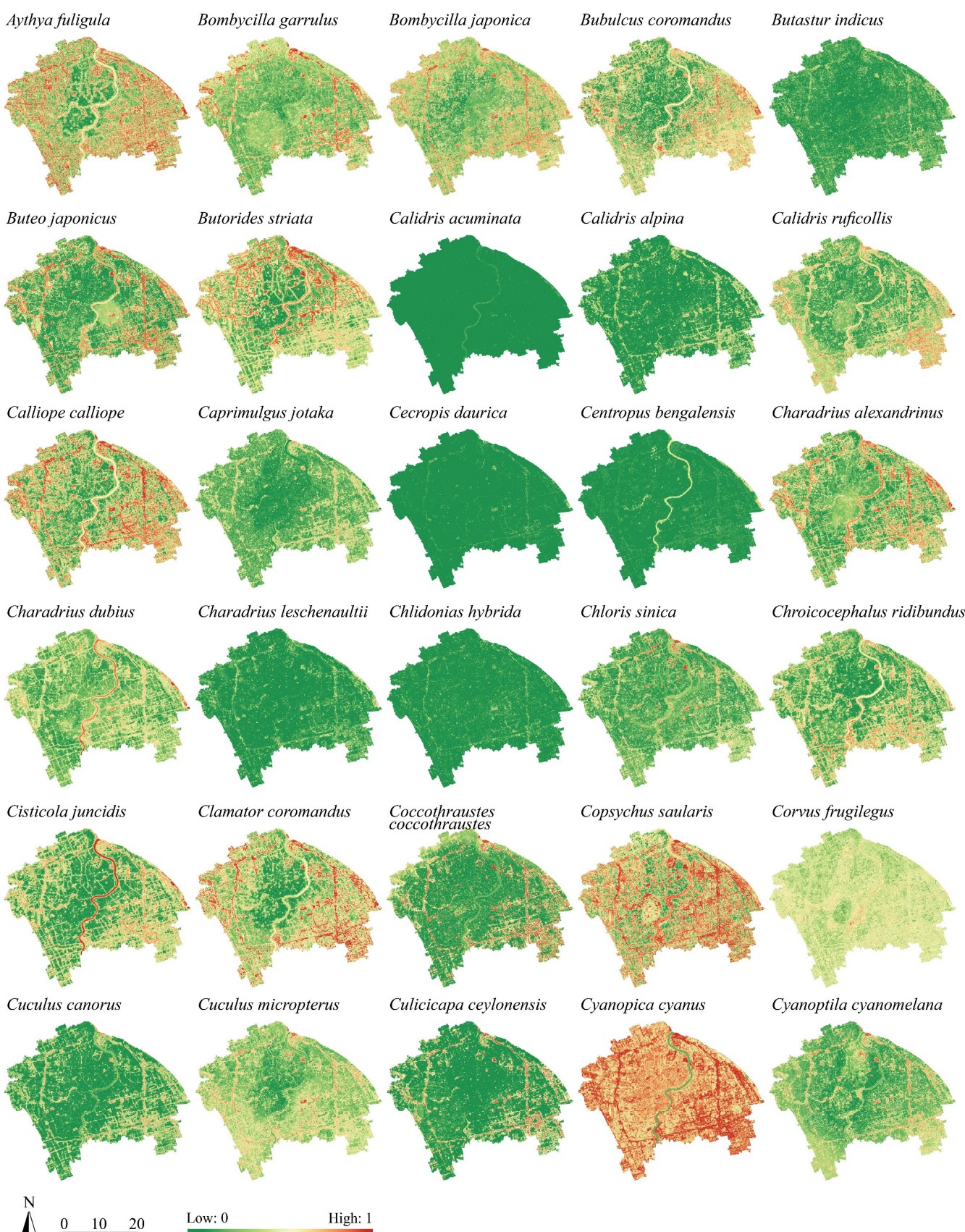

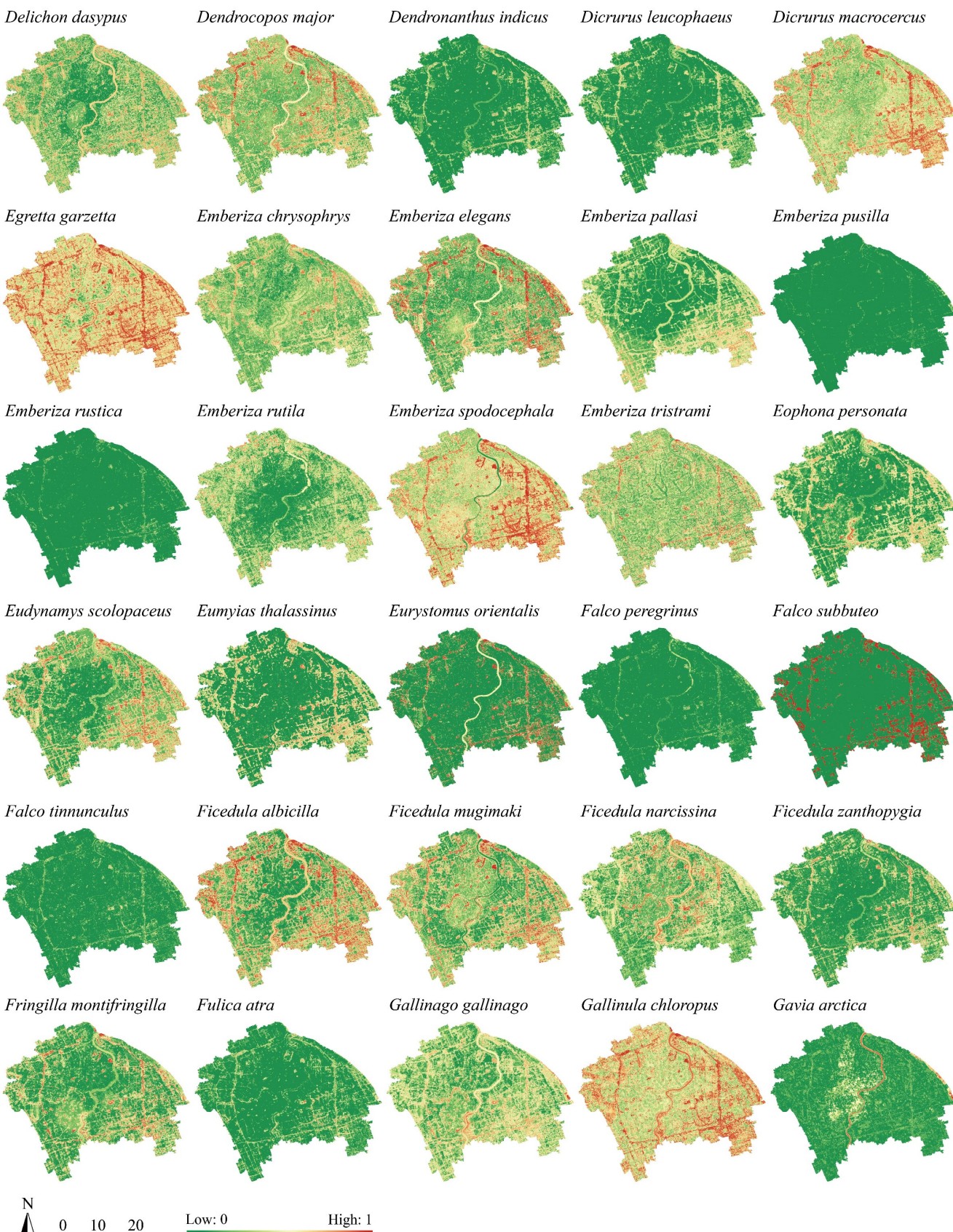

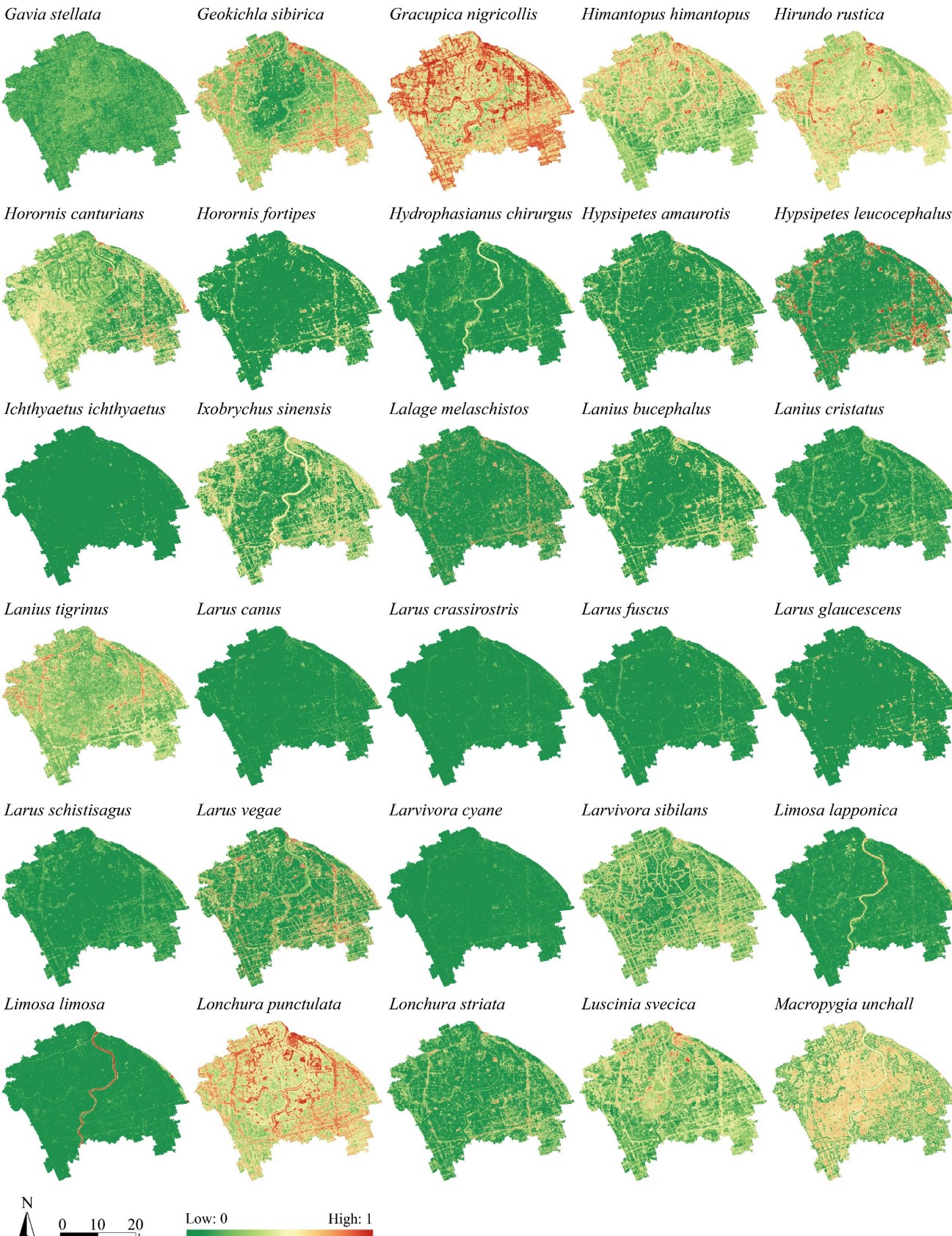

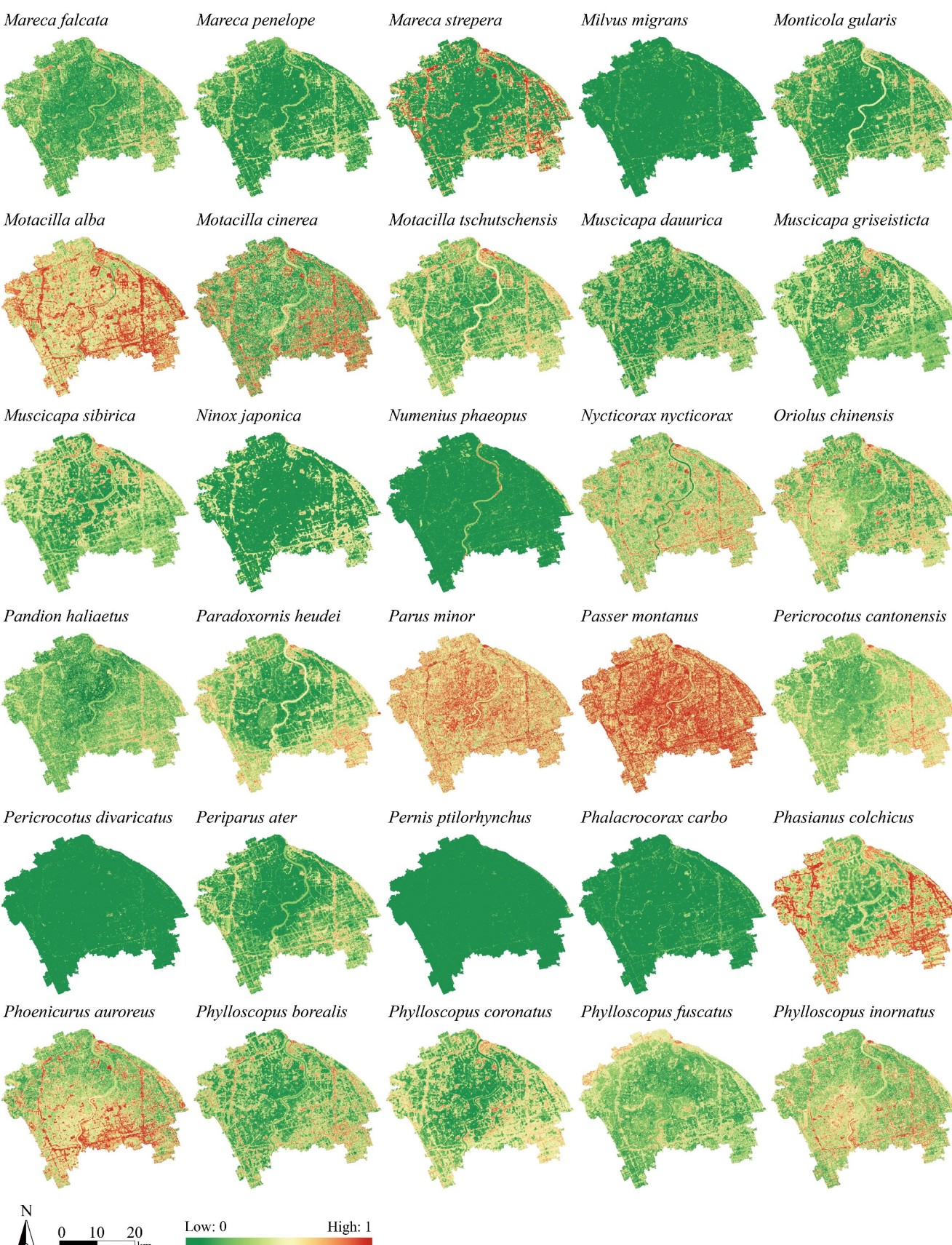

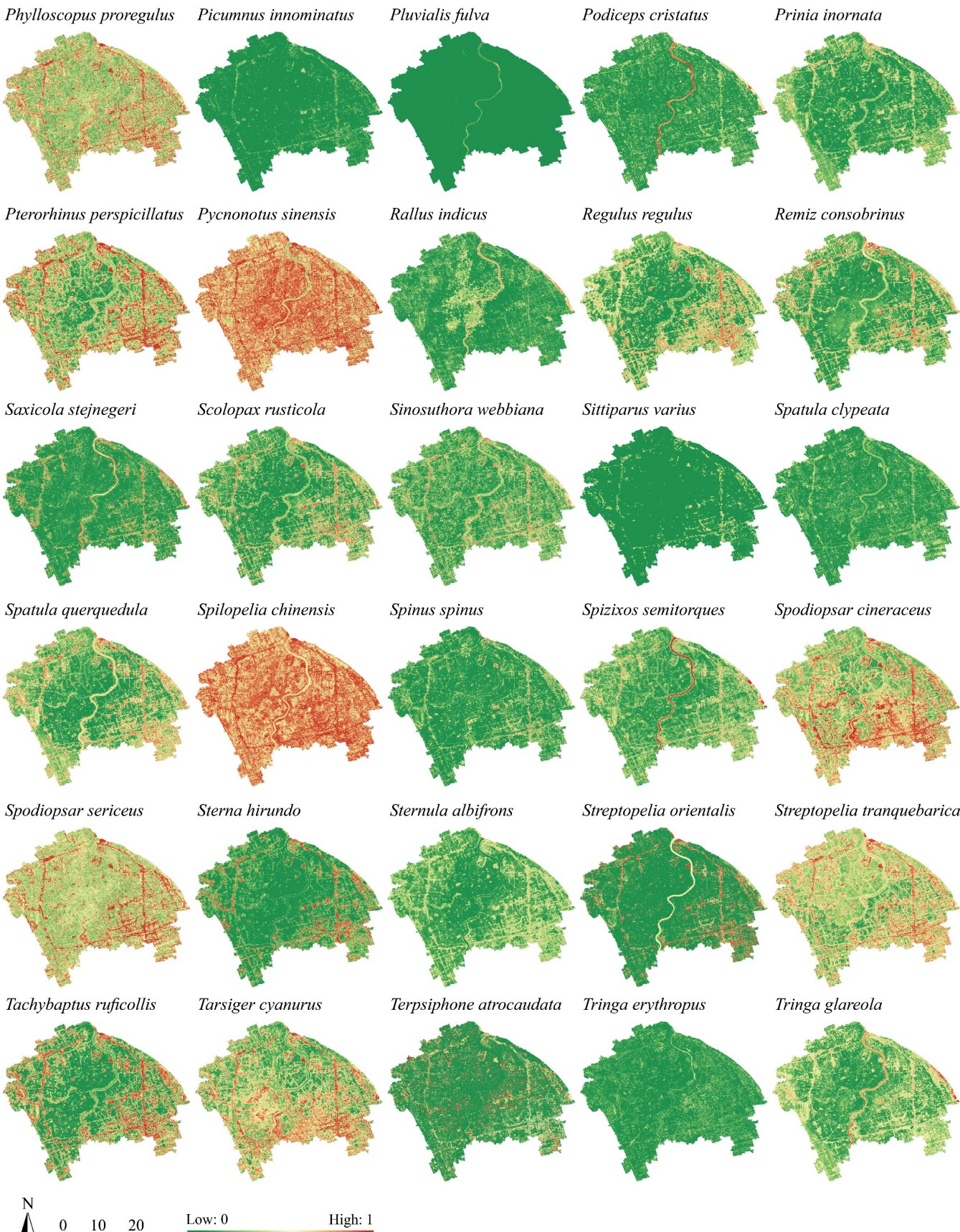

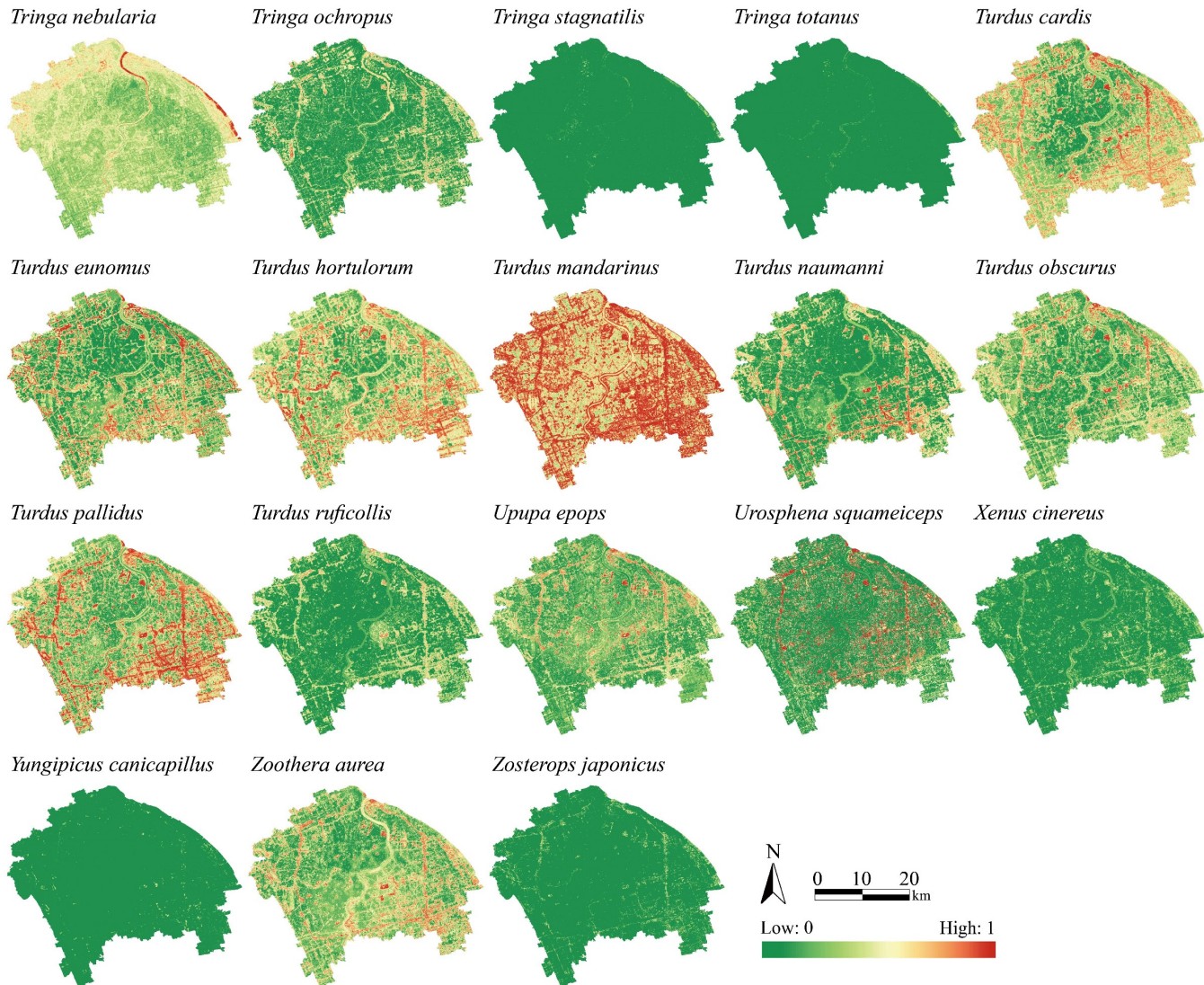

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
