# Peer review of "Identification of Bird Habitat Restoration Priorities in a Central Area of a Megacity"

_forests, doi:10.3390/f14081689_

Round 1
Reviewer 1 Report
This manuscript addresses interesting questions about the habitat restoration for birds in a megacity using citizen science. The statistical analysis used is appropriate to the objectives set out in the study, but some aspects of the methodology need to be better described.
Introduction
There is a lack of references of interest about the use of data from citizen science for SDMs, for example:
Arenas-Castro, S., Regos, A., Martins, I., Honrado, J. & Alonso, J. 2022. Effects of input data sources on species distribution model predictions across species with different distributional ranges. Journal of Biogeography, 49 (7): 1299-1312.
Robinson, O. J., Ruiz-Gutierrez, V., Reynolds, M. D., Golet, G. H., Strimas-Mackey, M. & Fink, D. 2020. Integrating citizen science data with expert surveys increases accuracy and spatial extent of species distribution models. Diversity and Distributions, 26 (8): 976-986.
Study Area and Data Pre-processing
Line 67. Authors should clarify what S.alba is.
Lines 97-98. These data need a bibliographic reference.
Lines 115-116. The “2019 Shanghai Bird List and the A Checklist on the Classification and Distribution of the Birds of China (Third Edition)” need a bibliographic reference.
Lines 116-118. Authors should better detail the sources of the observations used. For example the URL of China Bird Watching Record Center and the origin and location of the bird observation reports.
Lines 118-121. Authors allude to sampling with point counts and the consistency with the observations used. However, they do not provide information about the characteristics of this sampling. Where were those point counts located? What was the duration of each point count?. How to determine the congruence between the results of the point counts and the observations used in the modeling?. Merely asserting consistency is not acceptable.
Lines 122-123. It is necessary to indicate the period covered (years) by the observations used. It is also necessary to include as supplementary material at least the following information: species name, number of occurrences, phenology status and IUCN category.
Line 126. I suggest replacing "Environment Factors Selection" with "Environmental Predictors”.
Line 131. The authors must indicate the bibliographic source used to assign the phenological status of each species. This status should be included in the supplementary material.
Lines 142-143. Authors should adequately define the concept of transit birds. Are all species migratory? If so, why differentiate between transit birds and summer migratory or winter migratory?.
Discussion
Line 311. Section 5.1. has a dubious fit in the Discussion. Would fit better as a results section, although it should be shortened.
Author Response
Response to Reviewer 1 Comments
Dear Reviewer,
Thank you for your insightful comments on our manuscript. We greatly appreciate the value and guidance you provide for revising and improving our paper. Your comments have been carefully studied, and we have made the necessary corrections, indicated in red in the paper. We sincerely hope that these revisions meet with your approval.
Best regards.
General comments: This manuscript addresses interesting questions about the habitat restoration for birds in a megacity using citizen science. The statistical analysis used is appropriate to the objectives set out in the study, but some aspects of the methodology need to be better described.
Point 1: There is a lack of references of interest about the use of data from citizen science for SDMs, for example:
Arenas-Castro, S., Regos, A., Martins, I., Honrado, J. & Alonso, J. 2022. Effects of input data sources on species distribution model predictions across species with different distributional ranges. Journal of Biogeography, 49 (7): 1299-1312.
Robinson, O. J., Ruiz-Gutierrez, V., Reynolds, M. D., Golet, G. H., Strimas-Mackey, M. & Fink, D. 2020. Integrating citizen science data with expert surveys increases accuracy and spatial extent of species distribution models. Diversity and Distributions, 26 (8): 976-986.
Response 1: Thank you very much for the comment. We have incorporated a comprehensive review on the application of citizen science data in bird research in the introduction section. We have added the references you suggested and supplemented the reference with additional relevant sources. The results of the revision can be found below.
Revision: “Species Distribution Models (SDMs), also known as Ecological Niche Models (ENMs) [22-24], are used to model species distribution based on known distribution points and associated environmental factors. Simulating bird distribution at a large spatial scale requires a substantial quantity of bird data. However, traditional bird community surveys, employing structured methods such as point or transect counts [25, 26], entail significant time and manpower costs for collecting bird data. Consequently, these limitations impede large-scale research on bird diversity. The advancement of data collection technology has facilitated the study of bird diversity [27]. Previous studies by Sullivan et al. [28], Squires et al. [29], Liu et al. [30], and Wong et al. [31] have used data from bird observation websites, such as eBird, the China Bird Watching Record Center, iNaturalist, and Burungnesia, to analyze bird communities and detect bird diversity patterns at different scales. However, there is still limited research utilizing citizen science data in SDMs [32-34].” (Page 2 in no changes version; Page 2-3, Lines 92-104 in changes version)
[32] Dai, S.; Feng, D.; Xu, B. Monitoring potential geographical distribution of four wild bird species in China. Environmental Earth Sciences 2016, 75, doi:10.1007/s12665-016-5289-y.
[33] Robinson, O.J.; Ruiz‐Gutierrez, V.; Reynolds, M.D.; Golet, G.H.; Strimas‐Mackey, M.; Fink, D.; Maiorano, L. Integrating citizen science data with expert surveys increases accuracy and spatial extent of species distribution models. Diversity and Distributions 2020, 26, 976-986, doi:10.1111/ddi.13068.
[34] Arenas‐Castro, S.; Regos, A.; Martins, I.; Honrado, J.; Alonso, J. Effects of input data sources on species distribution model predictions across species with different distributional ranges. Journal of Biogeography 2022, 49, 1299-1312, doi:10.1111/jbi.14382.
Point 2: Line 67. Authors should clarify what S.alba is.
Response 2: We thank the reviewer for your suggestions. S.alba is the name of the study area mentioned in reference [18]. We have changed "S.alba" to its full name "Salix alba".
Revision: “From a habitat perspective, Ganatsas et al. [18] assessed the ecological status of the Salix alba floodplain forests in Kerkini National Park to maintain its support for bird diversity in the region.” (Page 2, Lines 67 in no changes version; Page 2, Line 71 in changes version)
Point 3: Lines 97-98. These data need a bibliographic reference.
Response 3: Thanks for your suggestion. We have added a bibliographic reference to support the data.
Revision: “As of 2019, Shanghai had a total of 494 bird species from 78 families and 22 orders, ac-counting for 33.51% of Chinese total bird species [36].” (Page 3, Lines 97-98 in no changes version; Page 3, Lines 115-116 in changes version)
[36] Guangmei, Z. A Checklist on the Classification and Distribution of the Birds of China (Third Edition); Science Press: Beijing, China, 2017, ISBN 978-7-03-054751-4
Point 4: Lines 115-116. The “2019 Shanghai Bird List and the A Checklist on the Classification and Distribution of the Birds of China (Third Edition)” need a bibliographic reference.
Response 4: Thanks for your advice. We have added bibliographic reference to the corresponding books and reports.
Revision: “In this study, a total of 494 potential bird species in Shanghai were selected based on the A Checklist on the Classification and Distribution of the Birds of China (Third Edition) [36] and the 2019 Shanghai Bird List [37].” (Page 3, Lines 115-116 in no changes version; Page 3, Lines 128-130 in changes version)
[36] Guangmei, Z. A Checklist on the Classification and Distribution of the Birds of China (Third Edition); Science Press: Beijing, China, 2017, ISBN 978-7-03-054751-4
[37] Shanghai Wild Bird Society. 2020. 2019 Shanghai Bird List. Shanghai: Shanghai Wild Bird Society.
Point 5: Lines 116-118. Authors should better detail the sources of the observations used. For example the URL of China Bird Watching Record Center and the origin and location of the bird observation reports.
Response 5: Thank you for your suggestions. We have provided detailed descriptions of the data sources, including the URLs of bird observation websites, the time range of bird observations, and the sources of bird observation reports.
Revision: “Bird observation data were collected from eBird (https://ebird.org/map) and the China Bird Watching Record Center (http://www.birdreport.cn/) for the period of 2010 to 2023 in central Shanghai. The birding reports provided by the China Bird Watching website, which included observational data from multiple individual reports, were combined to enhance the verification of observed quantities of bird species and to filter out duplicate observations. From this comprehensive data, a total of 17,461 observation records were selected, representing 311 bird species, which accounted for 62.96% of the total number of species recorded in 2019 Shanghai Bird List [37].” (Page 3, Lines 116-118 in no changes version; Page 3-4, Lines 130-138 in changes version)
Point 6: Lines 118-121. Authors allude to sampling with point counts and the consistency with the observations used. However, they do not provide information about the characteristics of this sampling. Where were those point counts located? What was the duration of each point count?. How to determine the congruence between the results of the point counts and the observations used in the modeling?. Merely asserting consistency is not acceptable.
Response 6: Thanks for your detailed suggestion. The purpose of selecting sample plots for bird surveys is to assess the quality of citizen science data. We have provided detailed explanations about the survey process, including information about the sample plots, the date range of bird observations, the spatial extent of the observations, and the time intervals. Furthermore, we have included a detailed report on the survey results in the supplementary materials, including Table S2 (Area and location information of 20 survey sites), S3 (List of observed bird species), and S4 (Comparison between observational data and citizen science data).
Revision: “To assess the quality of citizen science data, we selected 20 sites during the breeding season in May 2023 and employed a point count method for bird species data collection [25]. Between 8:00 and 18:00, we conducted bird counts within a 25-meter radius for 10 minutes at each sampling point, ensuring a minimum spacing of 150 meters between points [38]. Further details regarding the study sites and bird count results can be found in Supplementary Materials S2-3. To evaluate the consistency between field observations and citizen science data, we employed the species repetition rate. This rate indicates the proportion of species richness observed both in the field and through citizen science, relative to the total species richness observed across field and citizen science data at a specific site. The calculated species repetition rate was 0.78±0.14 (Table S4), demonstrating the suitability of citizen science data quality for SDMs.” (Page 3, Lines 118-121 in no changes version; Page 4, Lines 147-157 in changes version)
Table S2
Area and location information of 20 survey sites. The latitude and longitude information donates the coordinates of the central point of the site. (Supplementary material)
|
Site ID |
Site name |
Area(hm2) |
Latitude |
Longitude |
|
S1 |
Yangpu Park |
20.23 |
31.282476 |
121.531938 |
|
S2 |
Peace Park |
16.07 |
31.272752 |
121.498739 |
|
S3 |
Luxun Park |
22.74 |
31.273735 |
121.479042 |
|
S4 |
Quyang Park |
6.50 |
31.289065 |
121.482053 |
|
S5 |
Jingan Park |
2.75 |
31.223870 |
121.442182 |
|
S6 |
Yanzhong Plaza |
5.06 |
31.227490 |
121.470490 |
|
S7 |
Gucheng park |
3.37 |
31.230485 |
121.489350 |
|
S8 |
Xujiahui Park |
7.94 |
31.199265 |
121.437869 |
|
S9 |
Fuxing Park |
7.32 |
31.218942 |
121.464586 |
|
S10 |
Zhabei Park |
12.00 |
31.272295 |
121.455310 |
|
S11 |
Binjiang Park |
10.10 |
31.244795 |
121.497766 |
|
S12 |
Taipingqiao Park |
2.09 |
31.221378 |
121.472248 |
|
S13 |
Shanghai Cultural Square |
9.88 |
31.213912 |
121.458458 |
|
S14 |
Bright City Greenland |
4.17 |
31.246562 |
121.455960 |
|
S15 |
Renming Rd Greenland |
1.42 |
31.230780 |
121.484501 |
|
S16 |
Qinjian Chunyuan |
5.41 |
31.197335 |
121.489622 |
|
S17 |
Nanyuan Waterfront Green Park |
10.66 |
31.193703 |
121.471115 |
|
S18 |
Huangpu Park |
1.97 |
31.243866 |
121.486316 |
|
S19 |
Zhongshan East 2Rd Greenland |
2.26 |
31.230717 |
121.491275 |
|
S20 |
Nansuzhou Rd Greenland |
3.39 |
31.244422 |
121.484497 |
Table S3
List of observed bird species. * represents citizen science data of bird species. (Supplementary material)
|
Bird species observed |
Site ID |
|||||||||||||||||||
|
S1 |
S2 |
S3 |
S4 |
S5 |
S6 |
S7 |
S8 |
S9 |
S10 |
S11 |
S12 |
S13 |
S14 |
S15 |
S16 |
S17 |
S18 |
S19 |
S20 |
|
|
Accipiter trivirgatus |
|
|
2(1*) |
|
|
|
(2*) |
|
|
|
|
|
|
|
|
|
1(1*) |
|
|
|
|
Acridotheres cristatellus |
|
|
|
|
3(5*) |
4 |
5(12*) |
3(10*) |
2(8*) |
1(3*) |
5(8*) |
|
(1*) |
|
1(1*) |
1(1*) |
|
|
|
|
|
Acridotheres tristis |
|
|
2 |
3 |
|
|
|
1(1*) |
|
2 |
|
|
|
|
|
|
(2*) |
|
|
|
|
Alcedo atthis |
|
|
(2*) |
|
|
|
(3*) |
1(3*) |
(3*) |
|
|
|
|
|
|
|
|
|
|
|
|
Anas platyrhynchos |
|
|
|
|
|
(2*) |
|
|
|
|
|
|
|
|
|
|
|
|
|
|
|
Butorides striata |
|
|
|
|
|
|
(2*) |
|
|
|
|
|
|
|
|
|
|
|
|
|
|
Calliope calliope |
|
|
|
|
|
|
(1*) |
|
|
|
|
|
|
|
|
|
1(1*) |
|
|
|
|
Chloris sinica |
|
|
(1*) |
|
|
|
(3*) |
1(1*) |
3(3*) |
|
|
|
|
|
|
|
|
|
|
|
|
Copsychus saularis |
|
|
2(4*) |
|
5(5*) |
2(2*) |
5(12*) |
3(9*) |
2(7*) |
|
1(2*) |
|
|
|
|
|
1 |
|
|
|
|
Cyanopica cyanus |
1(1*) |
|
|
|
|
|
(4*) |
1(4*) |
|
|
1(5*) |
1(1*) |
|
|
|
|
3(1*) |
|
|
|
|
Cygnus atratus |
|
|
|
|
|
|
|
4 |
|
|
|
|
|
|
|
|
|
|
|
|
|
Egretta garzetta |
|
|
|
|
|
(3*) |
1(3*) |
1(4*) |
(2*) |
(1*) |
2(5*) |
|
|
|
|
|
2(1*) |
|
|
1(1*) |
|
Emberiza tristrami |
|
|
|
|
|
|
|
|
|
|
1 |
|
|
|
|
|
|
|
|
|
|
Falco tinnunculus |
|
|
(1*) |
|
|
|
|
|
|
|
|
|
|
|
|
|
|
|
|
|
|
Ficedula mugimaki |
|
|
6(1*) |
|
|
|
(3*) |
(3*) |
|
|
|
|
|
(1*) |
|
|
|
|
|
1(1*) |
|
Fringilla montifringilla |
|
|
|
|
|
|
(1*) |
|
|
(1*) |
|
|
|
1(1*) |
|
|
|
|
|
|
|
Garrulax canorus |
|
|
|
|
|
|
|
|
|
|
1 |
|
|
|
|
|
|
|
|
|
|
Lanius cristatus |
|
|
|
|
|
|
1(1*) |
|
|
|
|
|
|
|
|
|
|
|
|
|
|
Lonchura striata |
1(1*) |
|
5(1*) |
|
|
|
|
(4*) |
(3*) |
|
|
|
|
|
|
|
1(1*) |
|
|
|
|
Luscinia svecica |
|
|
|
|
|
|
|
(1*) |
|
|
|
|
|
|
|
|
|
|
|
|
|
Motacilla alba |
|
|
|
|
2(3*) |
3(5*) |
2(6*) |
2(8*) |
1(4*) |
(3*) |
1(3*) |
(1*) |
1 |
|
1(1*) |
|
(1*) |
|
1(1*) |
|
|
Muscicapa dauurica |
|
|
|
(1*) |
|
|
1(4*) |
|
1(6*) |
|
|
|
|
|
|
|
|
|
|
|
|
Nycticorax nycticorax |
(2*) |
|
1(1*) |
|
|
(2*) |
|
|
|
|
1(2*) |
|
|
|
|
|
2 |
|
|
|
|
Parus minor |
2(1*) |
|
3(4*) |
|
1(1*) |
|
3(7*) |
|
|
1(3*) |
2(3*) |
|
|
|
1 |
|
|
|
|
|
|
Passer montanus |
5 |
3 |
9(7*) |
12(10*) |
5(6*) |
3(2*) |
15(21*) |
11(18*) |
5(12*) |
7(2*) |
5 |
5(1*) |
3(2*) |
4(3*) |
5(8*) |
2(7*) |
8(6*) |
5(1*) |
3(1*) |
3(5*) |
|
Phoenicurus auroreus |
|
|
|
|
|
2(5*) |
1(6*) |
(3*) |
|
(2*) |
(2*) |
|
1(1*) |
|
|
|
|
|
|
|
|
Phoenicurus fuliginosus |
|
|
|
|
|
|
1(5*) |
|
|
|
1(3*) |
|
|
|
|
|
|
|
|
|
|
Phylloscopus borealis |
|
|
|
1(1*) |
|
|
|
(2*) |
1(3*) |
|
|
|
|
|
|
|
|
|
|
|
|
Phylloscopus inornatus |
|
|
|
(1*) |
|
|
1(3*) |
2(7*) |
1(5*) |
1(1*) |
1(1*) |
|
1(2*) |
|
2 |
|
(3*) |
|
1(1*) |
|
|
Phylloscopus proregulus |
|
|
2(3*) |
|
|
|
|
3(5*) |
|
|
|
|
|
|
(2*) |
|
|
|
|
|
|
Pterorhinus perspicillatus |
|
|
|
|
|
|
1(1*) |
|
|
|
|
|
|
|
|
|
|
|
|
|
|
Pycnonotus sinensis |
6(1*) |
2(2*) |
2(5*) |
2(1*) |
9(11*) |
7(3*) |
5(12*) |
18(31*) |
10(26*) |
10(3*) |
6(2*) |
3(5*) |
2(1*) |
6(2*) |
2(1*) |
2(1*) |
4(4*) |
3(3*) |
|
|
|
Sinosuthora webbiana |
5(1*) |
|
3(2*) |
|
|
|
|
|
|
(4*) |
2(4*) |
|
|
2 |
|
2(2*) |
1(1*) |
|
|
|
|
Spilopelia chinensis |
2(1*) |
3(2*) |
2(4*) |
2(1*) |
3(4*) |
4 |
6(18*) |
2(3*) |
3(7*) |
3(4*) |
4(4*) |
2(2*) |
5(1*) |
|
2(2*) |
|
3(4*) |
|
|
|
|
Streptopelia orientalis |
|
|
|
|
|
|
|
1(1*) |
|
|
|
|
1(1*) |
|
|
|
|
|
|
|
|
Tachybaptus ruficollis |
|
2(3*) |
2 |
1(1*) |
|
|
|
|
|
|
|
|
|
|
|
|
|
|
|
|
|
Turdus cardis |
|
|
|
|
|
|
|
|
|
1(2*) |
(1*) |
|
|
|
|
|
|
|
|
|
|
Turdus hortulorum |
|
|
(2*) |
|
|
|
|
1(1*) |
|
|
1(1*) |
|
|
|
|
|
|
|
|
|
|
Turdus mandarinus |
3(1*) |
1(1*) |
2(10*) |
3(1*) |
5(7*) |
4(4*) |
5(11*) |
10(18*) |
5(7*) |
1(1*) |
1(1*) |
3(2*) |
3(2*) |
3(2*) |
2(4*) |
1(1*) |
3(3*) |
|
|
|
|
Turdus obscurus |
|
|
2(1*) |
|
|
|
|
|
|
|
|
|
|
|
|
|
|
|
|
|
|
Turdus pallidus |
|
|
1(1*) |
|
|
1(1*) |
|
2(4*) |
1(3*) |
|
|
|
|
|
1(1*) |
|
|
|
|
|
|
Yuhina torqueola |
1 |
|
|
|
|
|
|
|
|
|
|
|
|
|
|
|
|
|
|
|
Table S4
Comparison between observational data and citizen science data. Species repetition rate refers to the ratio of the richness of species observed both in the field and through citizen science, to the total richness of species observed in the field and through citizen science in a given site. (Supplementary material)
|
Site ID |
Bird richness (observational data) |
Bird richness (citizen science data) |
Species repetition rate |
|
S1 |
9 |
8 |
0.70 |
|
S2 |
5 |
4 |
0.80 |
|
S3 |
16 |
18 |
0.70 |
|
S4 |
7 |
8 |
0.67 |
|
S5 |
8 |
8 |
1.00 |
|
S6 |
9 |
10 |
0.58 |
|
S7 |
15 |
23 |
0.65 |
|
S8 |
18 |
22 |
0.74 |
|
S9 |
12 |
15 |
0.80 |
|
S10 |
9 |
12 |
0.61 |
|
S11 |
17 |
17 |
0.74 |
|
S12 |
5 |
6 |
0.83 |
|
S13 |
8 |
8 |
0.78 |
|
S14 |
5 |
5 |
0.67 |
|
S15 |
9 |
8 |
0.70 |
|
S16 |
5 |
5 |
1.00 |
|
S17 |
12 |
13 |
0.67 |
|
S18 |
2 |
2 |
1.00 |
|
S19 |
3 |
3 |
1.00 |
|
S20 |
3 |
3 |
1.00 |
Point 7: Lines 122-123. It is necessary to indicate the period covered (years) by the observations used. It is also necessary to include as supplementary material at least the following information: species name, number of occurrences, phenology status and IUCN category.
Response 7: We are very grateful for the reviewer’s comment. We have supplemented the time range for acquiring avian data. Additionally, in Supplementary materials (Table S1), we have provided the frequency of occurrence, IUCN category, and phenological status for each bird species’ data.
Revision: “Bird observation data were collected from eBird (https://ebird.org/map) and the China Bird Watching Record Center (http://www.birdreport.cn/) for the period of 2010 to 2023 in central Shanghai.” (Page 3, Lines 122-123 in no changes version; Page 3, Lines 130-132 in changes version)
“Among these, four species were classified as Critically Endangered (CR), four as Endangered (EN), four as Vulnerable (VU), and fifteen as Near Threatened (NT) (see Table S1 in the supplementary materials for the IUCN categories of each species). In accordance with the classification proposed by Zheng et al. [36], birds were categorized into four groups: resident birds, summer migratory birds, winter migratory birds, and transient birds (Table 1). This classification served as the basis for determining the phenological status of bird species in Shanghai. Detailed information regarding the phenological status of each species in our study can be found in Table S1 of the supplementary materials.” (Page 3, Lines 120-125 in no changes version; Page 4, Lines 138-148 in changes version)
Table S1
A total of 311 bird species data were collected, and bird species with occurrence counts less than or equal to 5 were excluded when applying the RF model. The classification of phenological status was based on the method proposed by Zheng et al., which includes the following categories: RB (Resident birds), SMB (Summer migratory birds), WMB (Winter migratory birds), and TB (Transit birds). IUCN: International Union for Conservation of Nature, divided into five levels, LC(Least Concern); NT(Near Threatened); VU(Vulnerable); EN(Endangered); CR(Critically Endangered). (Supplementary material)
|
Bird species |
Number of occurrences |
IUCN category |
Phenological status |
|
Abroscopus albogularis |
19 |
LC |
RB |
|
Accipiter gentilis |
4 |
LC |
WMB |
|
Accipiter gularis |
8 |
LC |
WMB |
|
Accipiter nisus |
12 |
LC |
WMB |
|
Accipiter soloensis |
18 |
LC |
RB |
|
Accipiter trivirgatus |
48 |
LC |
RB |
|
Acridotheres cristatellus |
454 |
LC |
RB |
|
Acrocephalus bistrigiceps |
6 |
LC |
TB |
|
Acrocephalus orientalis |
12 |
LC |
SMB |
|
Actitis hypoleucos |
56 |
LC |
WMB |
|
Aegithalos concinnus |
81 |
LC |
RB |
|
Aegithalos glaucogularis |
58 |
LC |
RB |
|
Aerodramus brevirostris |
2 |
LC |
TB |
|
Aix galericulata |
70 |
LC |
WMB |
|
Alauda arvensis |
6 |
LC |
WMB |
|
Alauda gulgula |
14 |
LC |
RB |
|
Alcedo atthis |
154 |
LC |
RB |
|
Amaurornis phoenicurus |
50 |
LC |
RB |
|
Anas acuta |
10 |
LC |
WMB |
|
Anas platyrhynchos |
85 |
LC |
WMB |
|
Anas zonorhyncha |
205 |
LC |
WMB |
|
Anser fabalis |
2 |
LC |
WMB |
|
Anthus cervinus |
12 |
LC |
WMB |
|
Anthus gustavi |
4 |
LC |
TB |
|
Anthus hodgsoni |
221 |
LC |
WMB |
|
Anthus richardi |
18 |
LC |
SMB |
|
Anthus rubescens |
13 |
LC |
WMB |
|
Anthus spinoletta |
3 |
LC |
WMB |
|
Antigone vipio |
1 |
LC |
WMB |
|
Apus nipalensis |
11 |
LC |
SMB |
|
Ardea alba |
74 |
LC |
SMB |
|
Ardea cinerea |
155 |
LC |
RB |
|
Ardea intermedia |
36 |
LC |
SMB |
|
Ardea purpurea |
7 |
LC |
SMB |
|
Ardeola bacchus |
82 |
LC |
SMB |
|
Arenaria interpres |
1 |
LC |
TB |
|
Arundinax aedon |
1 |
LC |
TB |
|
Asio flammeus |
3 |
LC |
WMB |
|
Asio otus |
3 |
LC |
WMB |
|
Aythya baeri |
7 |
CR |
WMB |
|
Aythya ferina |
18 |
VU |
WMB |
|
Aythya fuligula |
104 |
LC |
WMB |
|
Aythya marila |
3 |
LC |
WMB |
|
Bombycilla garrulus |
17 |
LC |
WMB |
|
Bombycilla japonica |
15 |
NT |
WMB |
|
Botaurus stellaris |
5 |
LC |
WMB |
|
Bubulcus coromandus |
26 |
/ |
SMB |
|
Bucephala clangula |
1 |
LC |
WMB |
|
Butastur indicus |
9 |
LC |
TB |
|
Buteo japonicus |
39 |
LC |
WMB |
|
Butorides striata |
23 |
LC |
RB |
|
Calcarius lapponicus |
1 |
LC |
WMB |
|
Calidris acuminata |
7 |
LC |
TB |
|
Calidris alba |
4 |
LC |
TB |
|
Calidris alpina |
7 |
LC |
WMB |
|
Calidris canutus |
2 |
NT |
TB |
|
Calidris falcinellus |
2 |
LC |
TB |
|
Calidris minuta |
1 |
LC |
TB |
|
Calidris pugnax |
1 |
LC |
TB |
|
Calidris pygmea |
3 |
CR |
WMB |
|
Calidris ruficollis |
10 |
NT |
TB |
|
Calidris subminuta |
3 |
LC |
TB |
|
Calidris temminckii |
3 |
LC |
WMB |
|
Calidris tenuirostris |
3 |
EN |
TB |
|
Calliope calliope |
8 |
LC |
TB |
|
Caprimulgus jotaka |
23 |
LC |
SMB |
|
Cecropis daurica |
26 |
LC |
TB |
|
Centropus bengalensis |
6 |
LC |
RB |
|
Ceryle rudis |
5 |
LC |
RB |
|
Charadrius alexandrinus |
21 |
LC |
TB |
|
Charadrius dubius |
21 |
LC |
TB |
|
Charadrius leschenaultii |
8 |
LC |
TB |
|
Charadrius mongolus |
3 |
LC |
TB |
|
Charadrius veredus |
5 |
LC |
TB |
|
Chlidonias hybrida |
10 |
LC |
TB |
|
Chlidonias leucopterus |
5 |
LC |
WMB |
|
Chloris sinica |
157 |
LC |
RB |
|
Chroicocephalus ridibundus |
43 |
LC |
WMB |
|
Circus cyaneus |
2 |
LC |
WMB |
|
Circus melanoleucos |
1 |
LC |
WMB |
|
Circus spilonotus |
1 |
LC |
WMB |
|
Cisticola juncidis |
20 |
LC |
RB |
|
Clamator coromandus |
7 |
LC |
RB |
|
Coccothraustes coccothraustes |
20 |
LC |
WMB |
|
Copsychus saularis |
479 |
LC |
RB |
|
Corvus corone |
1 |
LC |
WMB |
|
Corvus frugilegus |
8 |
LC |
RB |
|
Corvus macrorhynchos |
1 |
LC |
RB |
|
Coturnix japonica |
2 |
NT |
WMB |
|
Cuculus canorus |
21 |
LC |
SMB |
|
Cuculus micropterus |
58 |
LC |
SMB |
|
Cuculus poliocephalus |
3 |
LC |
SMB |
|
Cuculus saturatus |
1 |
LC |
SMB |
|
Culicicapa ceylonensis |
10 |
LC |
SMB |
|
Cyanopica cyanus |
388 |
LC |
RB |
|
Cyanoptila cyanomelana |
42 |
LC |
TB |
|
Cygnus columbianus |
1 |
LC |
WMB |
|
Cygnus cygnus |
1 |
LC |
WMB |
|
Cygnus olor |
1 |
LC |
TB |
|
Delichon dasypus |
7 |
LC |
TB |
|
Dendrocitta formosae |
4 |
LC |
RB |
|
Dendrocopos major |
23 |
LC |
RB |
|
Dendronanthus indicus |
6 |
LC |
TB |
|
Dicrurus hottentottus |
3 |
LC |
SMB |
|
Dicrurus leucophaeus |
6 |
LC |
SMB |
|
Dicrurus macrocercus |
17 |
LC |
SMB |
|
Egretta eulophotes |
5 |
VU |
SMB |
|
Egretta garzetta |
432 |
LC |
SMB |
|
Elanus caeruleus |
3 |
LC |
RB |
|
Emberiza chrysophrys |
37 |
LC |
TB |
|
Emberiza cioides |
1 |
LC |
RB |
|
Emberiza elegans |
130 |
LC |
WMB |
|
Emberiza fucata |
5 |
LC |
TB |
|
Emberiza pallasi |
16 |
LC |
TB |
|
Emberiza pusilla |
17 |
LC |
WMB |
|
Emberiza rustica |
26 |
VU |
WMB |
|
Emberiza rutila |
7 |
LC |
TB |
|
Emberiza spodocephala |
250 |
LC |
WMB |
|
Emberiza tristrami |
113 |
LC |
WMB |
|
Emberiza yessoensis |
2 |
NT |
WMB |
|
Eophona personata |
43 |
LC |
WMB |
|
Eudynamys scolopaceus |
13 |
LC |
SMB |
|
Eumyias thalassinus |
17 |
LC |
SMB |
|
Eurystomus orientalis |
31 |
LC |
TB |
|
Falco amurensis |
5 |
LC |
WMB |
|
Falco columbarius |
1 |
LC |
WMB |
|
Falco peregrinus |
53 |
LC |
WMB |
|
Falco subbuteo |
8 |
LC |
SMB |
|
Falco tinnunculus |
76 |
LC |
WMB |
|
Ficedula albicilla |
18 |
LC |
TB |
|
Ficedula mugimaki |
77 |
LC |
TB |
|
Ficedula narcissina |
27 |
LC |
TB |
|
Ficedula zanthopygia |
30 |
LC |
SMB |
|
Fringilla montifringilla |
126 |
LC |
WMB |
|
Fulica atra |
202 |
LC |
WMB |
|
Gallinago gallinago |
30 |
LC |
WMB |
|
Gallinago megala |
2 |
LC |
TB |
|
Gallinago stenura |
5 |
LC |
TB |
|
Gallinula chloropus |
210 |
LC |
RB |
|
Gavia arctica |
15 |
LC |
WMB |
|
Gavia stellata |
6 |
LC |
WMB |
|
Gelochelidon nilotica |
2 |
LC |
RB |
|
Geokichla sibirica |
10 |
LC |
TB |
|
Glareola maldivarum |
5 |
LC |
TB |
|
Gracupica nigricollis |
67 |
LC |
RB |
|
Grus grus |
1 |
LC |
WMB |
|
Halcyon pileata |
3 |
LC |
SMB |
|
Haliaeetus albicilla |
1 |
LC |
WMB |
|
Helopsaltes ochotensis |
3 |
LC |
TB |
|
Helopsaltes pryeri |
4 |
NT |
WMB |
|
Hemixos castanonotus |
5 |
LC |
RB |
|
Hierococcyx hyperythrus |
3 |
LC |
WMB |
|
Himantopus himantopus |
12 |
LC |
TB |
|
Hirundapus caudacutus |
1 |
LC |
TB |
|
Hirundo rustica |
239 |
LC |
SMB |
|
Horornis canturians |
36 |
LC |
SMB |
|
Horornis fortipes |
10 |
LC |
RB |
|
Hydrophasianus chirurgus |
6 |
LC |
SMB |
|
Hydroprogne caspia |
3 |
LC |
SMB |
|
Hypothymis azurea |
3 |
LC |
RB |
|
Hypsipetes amaurotis |
7 |
LC |
TB |
|
Hypsipetes leucocephalus |
9 |
LC |
SMB |
|
Ichthyaetus ichthyaetus |
6 |
LC |
TB |
|
Ixobrychus cinnamomeus |
2 |
LC |
SMB |
|
Ixobrychus eurhythmus |
1 |
LC |
SMB |
|
Ixobrychus flavicollis |
3 |
LC |
SMB |
|
Ixobrychus sinensis |
39 |
LC |
SMB |
|
Jynx torquilla |
5 |
LC |
WMB |
|
Lalage melaschistos |
9 |
LC |
SMB |
|
Lanius bucephalus |
8 |
LC |
WMB |
|
Lanius cristatus |
48 |
LC |
SMB |
|
Lanius sphenocercus |
5 |
LC |
WMB |
|
Lanius tigrinus |
12 |
LC |
SMB |
|
Larus canus |
28 |
LC |
WMB |
|
Larus crassirostris |
20 |
LC |
WMB |
|
Larus fuscus |
41 |
LC |
WMB |
|
Larus glaucescens |
7 |
LC |
WMB |
|
Larus schistisagus |
16 |
LC |
WMB |
|
Larus vegae |
108 |
/ |
WMB |
|
Larvivora akahige |
2 |
LC |
TB |
|
Larvivora cyane |
10 |
LC |
TB |
|
Larvivora sibilans |
27 |
LC |
TB |
|
Leucogeranus leucogeranus |
1 |
CR |
WMB |
|
Limnodromus scolopaceus |
1 |
LC |
TB |
|
Limosa lapponica |
6 |
NT |
TB |
|
Limosa limosa |
6 |
NT |
TB |
|
Locustella lanceolata |
3 |
LC |
TB |
|
Lonchura punctulata |
42 |
LC |
RB |
|
Lonchura striata |
145 |
LC |
RB |
|
Luscinia svecica |
11 |
LC |
TB |
|
Macropygia unchall |
13 |
LC |
RB |
|
Mareca falcata |
27 |
NT |
WMB |
|
Mareca penelope |
7 |
LC |
WMB |
|
Mareca strepera |
51 |
LC |
WMB |
|
Mergellus albellus |
2 |
LC |
WMB |
|
Mergus merganser |
1 |
LC |
WMB |
|
Milvus migrans |
6 |
LC |
RB |
|
Monticola gularis |
7 |
LC |
TB |
|
Monticola solitarius |
5 |
LC |
RB |
|
Motacilla alba |
765 |
LC |
RB |
|
Motacilla cinerea |
73 |
LC |
TB |
|
Motacilla citreola |
1 |
LC |
WMB |
|
Motacilla tschutschensis |
42 |
LC |
TB |
|
Muscicapa dauurica |
76 |
LC |
TB |
|
Muscicapa griseisticta |
40 |
LC |
TB |
|
Muscicapa sibirica |
28 |
LC |
TB |
|
Myophonus caeruleus |
2 |
LC |
SMB |
|
Nettapus coromandelianus |
4 |
LC |
SMB |
|
Niltava davidi |
1 |
LC |
SMB |
|
Ninox japonica |
6 |
LC |
WMB |
|
Numenius arquata |
4 |
NT |
WMB |
|
Numenius madagascariensis |
5 |
EN |
TB |
|
Numenius minutus |
3 |
LC |
TB |
|
Numenius phaeopus |
7 |
LC |
TB |
|
Nycticorax nycticorax |
893 |
LC |
RB |
|
Onychoprion anaethetus |
1 |
LC |
SMB |
|
Oriolus chinensis |
26 |
LC |
SMB |
|
Pandion haliaetus |
7 |
LC |
RB |
|
Paradoxornis heudei |
12 |
NT |
RB |
|
Parus minor |
389 |
/ |
RB |
|
Passer montanus |
1145 |
LC |
RB |
|
Pelecanus crispus |
2 |
NT |
TB |
|
Pericrocotus cantonensis |
8 |
LC |
SMB |
|
Pericrocotus divaricatus |
16 |
LC |
TB |
|
Periparus ater |
6 |
LC |
RB |
|
Pernis ptilorhynchus |
7 |
LC |
TB |
|
Phalacrocorax carbo |
65 |
LC |
WMB |
|
Phalaropus lobatus |
5 |
LC |
TB |
|
Phasianus colchicus |
36 |
LC |
RB |
|
Phoenicurus auroreus |
285 |
LC |
WMB |
|
Phylloscopus borealis |
44 |
LC |
TB |
|
Phylloscopus castaniceps |
1 |
LC |
SMB |
|
Phylloscopus coronatus |
43 |
LC |
TB |
|
Phylloscopus fuscatus |
18 |
LC |
WMB |
|
Phylloscopus inornatus |
135 |
LC |
WMB |
|
Phylloscopus plumbeitarsus |
5 |
LC |
TB |
|
Phylloscopus proregulus |
256 |
LC |
WMB |
|
Picumnus innominatus |
19 |
LC |
RB |
|
Platalea leucorodia |
2 |
LC |
TB |
|
Platalea minor |
3 |
EN |
TB |
|
Plegadis falcinellus |
2 |
LC |
TB |
|
Pluvialis fulva |
9 |
LC |
TB |
|
Pluvialis squatarola |
2 |
LC |
TB |
|
Podiceps auritus |
1 |
VU |
WMB |
|
Podiceps cristatus |
29 |
LC |
WMB |
|
Podiceps nigricollis |
2 |
LC |
WMB |
|
Poecile palustris |
3 |
LC |
RB |
|
Prinia inornata |
61 |
LC |
RB |
|
Pterorhinus perspicillatus |
50 |
LC |
RB |
|
Pycnonotus sinensis |
748 |
LC |
RB |
|
Pycnonotus xanthorrhous |
1 |
LC |
RB |
|
Rallus indicus |
11 |
LC |
WMB |
|
Recurvirostra avosetta |
4 |
LC |
TB |
|
Regulus regulus |
41 |
LC |
WMB |
|
Remiz consobrinus |
30 |
LC |
WMB |
|
Riparia diluta |
2 |
LC |
RB |
|
Rostratula benghalensis |
2 |
LC |
RB |
|
Saxicola stejnegeri |
18 |
/ |
WMB |
|
Scolopax rusticola |
26 |
LC |
WMB |
|
Sinosuthora webbiana |
215 |
LC |
RB |
|
Sittiparus varius |
6 |
LC |
RB |
|
Spatula clypeata |
8 |
LC |
WMB |
|
Spatula querquedula |
7 |
LC |
TB |
|
Spilopelia chinensis |
817 |
LC |
RB |
|
Spinus spinus |
134 |
LC |
WMB |
|
Spizixos semitorques |
21 |
LC |
RB |
|
Spodiopsar cineraceus |
352 |
LC |
WMB |
|
Spodiopsar sericeus |
247 |
LC |
RB |
|
Sterna hirundo |
20 |
LC |
TB |
|
Sternula albifrons |
6 |
LC |
SMB |
|
Streptopelia orientalis |
175 |
LC |
RB |
|
Streptopelia tranquebarica |
15 |
LC |
RB |
|
Sturnus vulgaris |
2 |
LC |
TB |
|
Tachybaptus ruficollis |
199 |
LC |
RB |
|
Tadorna ferruginea |
1 |
LC |
WMB |
|
Tadorna tadorna |
3 |
LC |
WMB |
|
Tarsiger cyanurus |
307 |
LC |
WMB |
|
Terpsiphone atrocaudata |
10 |
NT |
TB |
|
Terpsiphone incei |
3 |
LC |
SMB |
|
Thalasseus bernsteini |
5 |
CR |
SMB |
|
Treron sieboldii |
4 |
LC |
TB |
|
Tringa brevipes |
3 |
NT |
TB |
|
Tringa erythropus |
6 |
LC |
TB |
|
Tringa glareola |
14 |
LC |
TB |
|
Tringa guttifer |
1 |
EN |
TB |
|
Tringa nebularia |
12 |
LC |
WMB |
|
Tringa ochropus |
24 |
LC |
WMB |
|
Tringa stagnatilis |
9 |
LC |
TB |
|
Tringa totanus |
8 |
LC |
TB |
|
Troglodytes troglodytes |
2 |
LC |
RB |
|
Turdus cardis |
25 |
LC |
TB |
|
Turdus chrysolaus |
4 |
LC |
SMB |
|
Turdus eunomus |
136 |
LC |
WMB |
|
Turdus hortulorum |
175 |
LC |
WMB |
|
Turdus mandarinus |
1293 |
LC |
RB |
|
Turdus mupinensis |
2 |
LC |
RB |
|
Turdus naumanni |
57 |
LC |
WMB |
|
Turdus obscurus |
51 |
LC |
TB |
|
Turdus pallidus |
262 |
LC |
WMB |
|
Turdus ruficollis |
7 |
LC |
WMB |
|
Turnix tanki |
1 |
LC |
SMB |
|
Upupa epops |
125 |
LC |
RB |
|
Urosphena squameiceps |
15 |
LC |
TB |
|
Vanellus cinereus |
4 |
LC |
SMB |
|
Vanellus vanellus |
4 |
NT |
WMB |
|
Xenus cinereus |
8 |
LC |
TB |
|
Yuhina torqueola |
1 |
LC |
RB |
|
Yungipicus canicapillus |
6 |
LC |
RB |
|
Zoothera aurea |
99 |
LC |
WMB |
|
Zosterops erythropleurus |
5 |
LC |
TB |
|
Zosterops japonicus |
8 |
LC |
SMB |
Point 8: Line 126. I suggest replacing “Environment Factors Selection” with “Environmental Predictors”.
Response 8: We have changed "Environment Factors Selection" to "Environmental Predictors."
Revision: “2.3. Environmental Predictors” (Page 3, Line 126 in no changes version; Page 4, Line 159 in changes version)
Point 9: Line 131. The authors must indicate the bibliographic source used to assign the phenological status of each species. This status should be included in the supplementary material.
Response 9: Thank you very much for the comment. We have added the reference for the classification of bird phenological states, and we have indicated the phenological status of each bird species in supplementary materials (Table S1).
Revision: “In accordance with the classification proposed by Zheng et al. [36], birds were categorized into four groups: resident birds, summer migratory birds, winter migratory birds, and transient birds (Table 1). This classification served as the basis for determining the phenological status of bird species in Shanghai.” (Page 4, Line 131 in no changes version; Page 4, Lines 141-144 in changes version)
[36] Guangmei, Z. A Checklist on the Classification and Distribution of the Birds of China (Third Edition); Science Press: Beijing, China, 2017, ISBN 978-7-03-054751-4
Point 10: Lines 142-143. Authors should adequately define the concept of transit birds. Are all species migratory? If so, why differentiate between transit birds and summer migratory or winter migratory?.
Response 10: We are very grateful for the reviewer’s comment. We have cited the definitions of each phenological state from reference [36], and in the paper, we have explained that the reason for distinguishing avian phenological states is to make the results of species distribution models (SDMs) more accurate. The specific reasons for distinguishing avian phenological states are as follows.
Revision: “In order to improve the accuracy of SDMs, we adopted a differentiated approach in selecting environmental factors, taking into account their temporal variability, particularly in relation to Bio1 (precipitation) and Bio2 (temperature). Therefore, we considered the phenological state of bird species when choosing the environmental factor. For resident and transit birds, annual precipitation and mean temperature were selected as factors. For summer and winter migratory birds, precipitation and mean temperature in summer and winter were selected as factors, respectively.” (Page 5, Lines 142-143 in no changes version; Page 4, Lines 163-169 in changes version)
Table 1. Definition of bird phenological status. (Page 4, Lines 158 in changes version)
|
Phenological status |
Definition |
|
Resident birds |
Birds that reside year-round within their habitat are collectively referred to as resident birds |
|
Summer migratory birds |
Migratory birds that breed in a specific region during summer, migrate to warmer southern regions for winter, and return to the same region for breeding the following spring are referred to as summer migratory birds in that particular area |
|
Winter migratory birds |
Birds that winter in a specific region, fly north for breeding in the following spring, and return to the same region for wintering in the autumn are referred to as winter migratory birds in that particular area |
|
Transit birds |
Birds that pass through a specific area during migration but do not breed or winter in that area are referred to as transient birds in that particular region |
Point 11: Line 311. Section 5.1. has a dubious fit in the Discussion. Would fit better as a results section, although it should be shortened.
Response 11: Thank you for your valuable suggestions. We have moved the text and charts regarding the results of section 5.1 to “4 Results - 4.4 Spatial Matching Types of Bird Richness and Habitat Quality”. Additionally, we have revised the discussion section to focus on the implications for bird habitat restoration.
Revision: 4.4 Spatial Matching Types of Bird Richness and Habitat Quality
“The standardized relationship between bird richness and habitat quality (Figure 9) shows distinct patterns. Forest and farmland display the highest bird richness in this type, with a median value of -0.64. Grassland and water body follow closely with a median value of -0.68. Conversely, urban construction land exhibits the lowest bird richness, scoring a median value of -0.88. In terms of habitat quality, water body ranks highest within this type, boasting a median value of -0.15, followed by forest (-0.18), grassland (-0.19), and farmland (-0.76). Urban construction land, on the other hand, presents the lowest habitat quality, with a median value of -1.60.” (Page 13, Lines 338-345 in changes version)
According to Figure 9, the results suggest that forest exhibits the highest levels of bird richness and habitat quality within this category, with median values of 0.98 and 0.94, respectively. Water body ranks second, with median values of 0.73 for bird richness and 0.78 for habitat quality. In contrast, grassland exhibits the lowest bird richness and comparatively lower habitat quality, with median values of 0.65 and 0.02, respectively.” (Page 13, Lines 350-355 in changes version)
Figure 9 illustrates that within this type, forest and grassland exhibit the highest bird richness, with a median value of -0.59. In contrast, water bodies demonstrate relatively lower bird richness, with a median value of -0.72. Additionally, forest stands out as having the highest habitat quality (0.87), followed by water bodies (0.77), while grassland shows comparatively lower habitat quality, with a median value of 0.02.” (Page 13, Lines 360-365 in changes version)
Based on Figure 9, the results indicate that forest has the highest bird richness within this type, with a median value of 0.73. It is closely followed by grassland (0.65), urban construction land (0.52), farmland (0.44), and water body (0.40). Regarding habitat quality, water body exhibits relatively higher quality with a median value of -0.14, followed by grassland (-0.15), forest (-0.17), and farmland (-0.74). Notably, urban construction land displays the lowest habitat quality, with a median value of -1.60.” (Page 13-14, Lines 371-377 in changes version)
5.2. Implications for Bird Habitat Restoration
“The HBR-LHQ area, primarily consisting of grassland and urban construction land, is a vital habitat in need of restoration. The coexistence of natural and built environments in this area contributes to its low habitat quality and high bird richness. This coexistence facilitates diverse habitat types and food sources, benefiting urban adapters and urban exploiters, ultimately leading to an increase in bird richness [60]. However, human disturbances pose a significant threat to this habitat type, resulting in reduced habitat quality. Consequently, restoring the regional habitat becomes imperative to enhance bird habitat suitability. To achieve this, it is recommended to prioritize the preservation of semi-natural vegetation and maintain the quality of existing forest. Simultaneously, afforestation efforts should be directed towards unused and abandoned lands to increase vegetation coverage and provide additional bird habitats.
The LBR-LHQ area, encompassing more than half of the urban construction land, presents a challenge for bird habitation due to relatively low habitat quality despite high bird richness in the forest, grassland, and water body habitats. The fragmented nature of these ecological patches necessitates the establishment of ecological corridors to connect them and create a cohesive network of bird habitats within the region. Moreover, optimizing the sizes of forest and grassland areas in the urban landscape and carefully considering their proximity to water sources and other landscape elements are crucial steps. These measures are essential for enhancing the suitability of bird habitats, increasing their chances of survival in urban areas, and ultimately augmenting bird richness.
The LBR-HHQ area primarily consists of forest and water body. While the water body exhibits relatively high habitat quality, it lacks suitable nesting sites for birds, resulting in lower bird richness. On the other hand, the forest exhibits higher habitat quality, albeit still lower compared to the HBQ-HHQ region. To enhance bird richness in this region, a transformation into a composite wetland habitat can be achieved by surrounding the core water body with forest, grassland, and reeds. This integration and connectivity of diverse landscape elements within the habitat will provide birds with varied foraging and nesting conditions, thereby augmenting bird richness.
The HBR-HHQ area represents a well-coordinated ecosystem, predominantly characterized by high-quality forest and grassland habitats. Notably, the forest and water body within this region exhibit high bird richness. As there is no urban construction land or farmland in this area, it presents favorable conditions for bird survival, emphasizing the need for habitat conservation strategies. City managers should prioritize the protection of forest and bird habitats along riverbanks. Optimizing vegetation community structure, enhancing vegetation diversity, and increasing spatial complexity are recommended measures to sustain the area's high bird richness status.” (Page 14-15, Lines 404-439 in changes version)
Reviewer 2 Report
The authors proposed and implemented a research framework for assessing priority habitats for birds in the core area of Shanghai. Their idea is interesting and innovative and can provide critical insights for the conservation and enhancement of the bird community of Shanghai. It can also serve as a paradigm for other similar areas.
Comments
Lines 2-4 – I would urge the authors to modify the article for brevity as “Identification of bird habitat restoration priorities in a central area of a megacity”.
Lines 25-26 – This sentence is not clear as it is written. I take it that approximately 50% of the forest habitat in shanghai was identified as critical for birds. Please rephrase.
Line 29 – Please define acronyms.
Lines 38039 – Also give the urbanization trend for China.
Lines 61-64 – Be careful not to refer to abundance when you talk about richness. Rephrase.
Line 91 – Accurately delineate…
Lines 99-103 – Please explain what biodiversity index is.
Lines 107-108, Figure 1 – Maps are numbered a, c, d. Please correct.
Lines 122-123 – 62.96% of the total number of species of China? Please be specific.
Line 125 – Near Threatened conservation status.
Line 126 – Environmental…
Table 1 – Define SHDI, NDVI.
Line 150 – Give reference for the InVEST model.
Lines 170, 187, 196 – Give references for R packages and software.
Line 210 – Tables 2 and 3…
Lines 253-255 – Not entirely true. It is the urban exploiters habitat of choice. A mosaic of green and grey habitats might be more diverse. But this id discussion.
Table 4 – Replace “Region” with “Spatial type” in the header of the first column.
Lines 312-313 – This is an awkward opening of the discussion. I also cannot see its use. Rather remove.
Lines 313-320 – The high bird richness in the low quality habitat of the HBR-LHQ spatial type could be due to high participation of built habitat. It could be that the presence of both natural and built environments favor urban adapters and urban exploiters thus increasing total richness. Please check and elaborate on this.
Lines 313-343 – Do not include results in Discussion section. Move Figure 8 and relevant text describing results to the Results section. Discuss the meaning of findings here, without giving results or referring to Figures and Tables.
Lines 347-380 – Same here. Move Figure 9 and relevant text describing results to the Results section. Discuss the meaning of findings here, without giving results or referring to Figures and Tables.
Author Response
Dear expert, thanks for your suggestions for the manuscript! Please see the attachment about the response.

Round 2
Reviewer 1 Report
The manuscript has adequately incorporated the suggested changes